# Position-dependent plasticity of distinct progenitor types in the primitive streak

**Filip J Wymeersch, Yali Huang, Guillaume Blin, Noemí Cambray, Ron Wilkie, Frederick CK Wong, Valerie Wilson\***

MRC Centre for Regenerative Medicine, Institute for Stem Cell Research, School of Biological Sciences, University of Edinburgh, Edinburgh, United Kingdom

**Abstract** The rostrocaudal (head-to-tail) axis is supplied by populations of progenitors at the caudal end of the embryo. Despite recent advances characterising one of these populations, the neuromesodermal progenitors, their nature and relationship to other populations remains unclear. Here we show that neuromesodermal progenitors are a single $Sox2^{low}T^{low}$ entity whose choice of neural or mesodermal fate is dictated by their position in the progenitor region. The choice of mesoderm fate is Wnt/β-catenin dependent. Wnt/β-catenin signalling is also required for a previously unrecognised phase of progenitor expansion during mid-trunk formation. Lateral/ventral mesoderm progenitors represent a distinct committed state that is unable to differentiate to neural fates, even upon overexpression of the neural transcription factor Sox2. They do not require Wnt/β-catenin signalling for mesoderm differentiation. This information aids the correct interpretation of *in vivo* genetic studies and the development of in vitro protocols for generating physiologically-relevant cell populations of clinical interest.

**\*For correspondence:** v.wilson@ ed.ac.uk

**Competing interests:** The authors declare that no competing interests exist.

## Introduction

The vertebrate rostrocaudal axis elongates in a rostral-to-caudal sequence by virtue of a population of progenitors at the caudal end of the embryo, in and near the primitive streak (PS) and later in the tail bud (TB; reviewed in *Wilson et al. (2009)*). Clonal analysis has demonstrated the presence of axial progenitors that behave as dual-fated tissue stem cells of the neurectoderm and somitic mesoderm (*Tzouanacou et al., 2009*), termed neuromesodermal progenitors (NMPs). Population fate maps at early somite stages have identified two regions of dual neurectoderm and mesoderm fate: the dorsal layer of the bilayered Node-Streak Border (NSB) and the Caudal Lateral Epiblast (CLE) on either side of the PS. The NSB also gives rise to the majority of the neuromesoderm-fated chordo-neural hinge (CNH) in the tail bud (*Cambray and Wilson, 2002*; *2007*). The adjacent midline PS and rostral node (RN) are fated exclusively for mesoderm and notochord, respectively (*Figure 1A–D*). The caudal-most tip of the CLE, and the adjoining caudal primitive streak, are solely fated for lateral/ventral mesoderm (LVM) (*Cambray and Wilson, 2007*). Thus cell fate in and around the PS is highly regionalised.

Fate mapping studies in early somite-stage mouse and chick embryos indicate a rostrocaudal organisation of mesoderm progenitors within the PS. Midline mesoderm is produced by the rostral PS, while successively more caudal regions of the streak are fated for the paraxial and lateral mesoderm (*Cambray and Wilson, 2007*; *Brown and Storey, 2000*; *Catala et al., 1996*; *Schoenwolf, 1992*; *Wilson and Beddington, 1996*). We have previously shown that cells fated exclusively for the medial somite in the NSB can be re-fated to more lateral regions of the somite on transplantation to the anterior primitive streak (*Cambray and Wilson, 2007*), indicating a degree of plasticity in paraxial mesoderm (PXM) fates. However, the extent of mesoderm plasticity in the PS and CLE

**eLife digest** Our bodies, like those of all animals with a backbone, form during embryo development in a head-to-tail sequence. This process is fuelled by populations of proliferating cells called progenitor cells, which are found in an early embryonic structure called the primitive streak, and later at the tail-end of the embryo.

One of these populations – known as the neuromesodermal progenitors (or NMPs) – produces the animal's spinal cord, muscle and bone tissue. However, it is not clear how this cell population is maintained or what triggers these cells to specialise into the correct cell type. It is even unclear whether NMPs are a single cell type or a collection of several types of progenitor, each with a slightly different propensity to make spinal cord or muscle and bone. Answering these questions could inform the future development of cell-replacement therapies for conditions such as spinal injuries.

Wymeersch et al. used a range of techniques to identify, map the fate, and assess the developmental potential of progenitors in the primitive streak. This revealed fine-grained differences in the fates adopted by cells in the progenitor region. However, these regional differences were found to result from the progenitor cells' extensive ability to respond to signals they receive from their environment, rather than being hard-wired into the progenitor cells. In fact, Wymeersch et al. detected only two distinct cell types: the NMPs and a new cell population termed lateral/paraxial mesoderm progenitors (or LPMPs), which, unlike NMPs, do not form nerve cells.

Further experiments investigated the molecular signals present in the environment of these progenitors that help to decide their fate. NMPs respond to an important developmental signal, called Wnt, by adopting a so-called mesoderm fate. This signal also induces NMPs to undergo a previously unknown phase of proliferation during the formation of the animal's body. LPMPs, on the other hand, do not require Wnt to form mesoderm.

These findings show that studies with embryos can identify new progenitor populations that might be clinically relevant, and reveal new ways in which a cell's environment inside an embryo can determine its fate.

has not been fully investigated. In particular, it is not known whether axial, paraxial and lateral mesoderm progenitors in the PS region are committed to these fates.

Fate maps of the CLE in chick indicate that cells at the rostral or lateral edges of the CLE form more neurectoderm than those at caudal or medial positions, which tend to generate mesoderm (*Brown and Storey, 2000*; *Iimura et al., 2007*). Furthermore, fate maps of mouse and chick have suggested that CLE progenitors are more transitory than those in the NSB, since they do not contribute to long axial stretches or extensively to the CNH (*Cambray and Wilson, 2007*; *Brown and Storey, 2000*; *Iimura et al., 2007*). However, the potency of these populations has not been tested, leaving open the question of whether these regional differences are simply the product of differing cellular environments acting on a single cell type to determine fate, or whether multiple populations with restricted potency exist within the PS region.

*T, Tbx6, Fgfr1* and *Wnt3a* are expressed in the PS region and required for correct mesoderm production, and loss of each of them leads both to shortened axes, and the ectopic production of neural tissue at the expense of somitic mesoderm (*Chapman and Papaioannou, 1998*; *Yamaguchi et al., 1999*; *Yoshikawa et al., 1997*; *Ciruna et al., 1997*). This suggests that NMP maintenance is intimately linked with preserving a balance between neurectoderm and mesoderm production. *Tbx6* expression in the midline PS represses *Sox2* in mesoderm-fated cells, ensuring suppression of the neural transcription program (*Takemoto et al., 2011*). Furthermore, in zebrafish, Wnt/β-catenin activation influences the decision of cells in both gastrula- and somite-stage embryos to enter neural or mesodermal lineages (*Martin and Kimelman, 2012*). More recently, lineage-tracing experiments showed that conditional deletion of Wnt3a or β-catenin in the T[+] cell compartment leads to a switch of primitive streak progenitors towards a neural fate (*Garriock et al., 2015*). However, constitutive Wnt/β-catenin activity in the T[+] cell compartment is not sufficient to divert all neural progenitors to mesoderm fates: providing cells in the caudal progenitor region with a stabilised

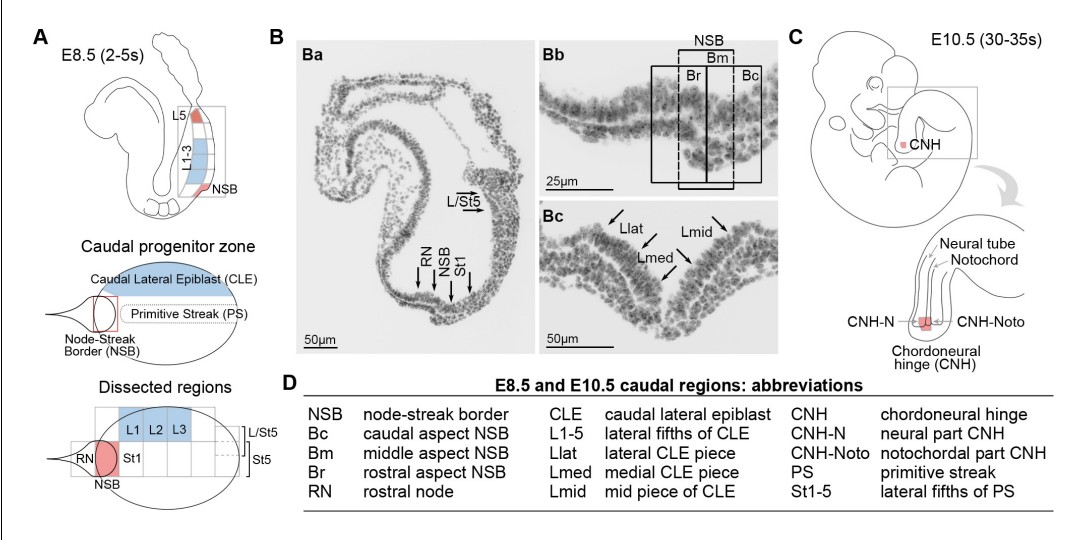

**Figure 1.** Location of neuromesodermal and lateral/ventral mesoderm progenitors. (**A**) Diagrams of the E8.5 (2–5 s) embryo showing the location of neuromesodermal and lateral/ventral-fated progenitors and the terminology used in this study. NSB, node-streak border; CLE, caudal lateral epiblast; RN, rostral node; St1, rostral 1/5 of the streak; St5, caudal 1/5 of the streak; L1-5, lateral fifths of the lateral epiblast, with L1 corresponding to most rostral adjacent to St1, and L5 adjacent to St5. (**B**) DAPI-stained sections through the E8.5 (2–5 s) embryo. (**Ba**) Transverse section illustrates dissected midline regions. (**Bb**) Magnified view of the NSB region shows the rostral, mid and caudal border (termed Br, Bm and Bc). (**Bc**) Transverse section through the mid primitive streak shows the position of Lmid, Llat and Lmed (mid, lateral and medial CLE, respectively). (**C**) At mid-gestation, NMPs are located in the chordoneural hinge (CNH). The CNH is composed of two parts: the dorsal (neural) part (termed CNH-N) and the ventral (notochordal) part (CNH-Noto). (**D**) Abbreviations of embryonic regions used in this manuscript.

form of β-catenin results in an enlarged PSM domain, but does not lead to loss of neural cell production (*Aulehla et al., 2008*; *Jurberg et al., 2014*). Moreover, enhanced β-catenin activity does not necessarily compromise the presence of NMPs in the CLE (*Garriock et al., 2015*). While these experiments point to an important role of Wnt signalling in axial progenitors, the promoters used do not specifically target NMPs. Grafting of precise NMP areas can provide a complementary approach that allows a direct assessment of the currently unresolved roles of Wnt signalling in NMPs and the caudal-most CLE.

In this study, we investigate the heterogeneity, plasticity and commitment of NMPs and lateral/ventral mesoderm progenitors, and the mechanisms by which they choose between alternative fates. We find that NMPs are committed to neuromesodermal lineages and choose between retention as progenitors, and differentiation as either neurectoderm or mesoderm based on their location within the progenitor region; the latter choice is β-catenin dependent. We show that NMPs express low levels of T and Sox2, and that during mid-trunk formation, Wnt/β-catenin signalling expands the number of Sox2$^+$T$^+$ NMPs and maintains the appropriate level of T in the NMP population. We further show that lateral/ventral mesoderm progenitors are exclusively mesoderm-committed yet show plasticity within the mesoderm lineage, and respond to distinct signalling and transcription factor cues from those that govern NMPs.

## Results

### Potency of NM-fated regions is restricted to neural and mesodermal lineages

The potency of NM-fated (NSB, L1-3, CNH) and surrounding regions was examined by transplantation under the kidney capsule (*Figure 2A–C*). Control grafts of embryonic day (E) 7.5 anterior (rostral) or posterior (caudal, PS-containing) parts of the late-streak or early headfold stage embryo formed large teratocarcinomas containing embryonal carcinoma (EC) cells and derivatives of all three germ layers including neural and non-neural ectoderm (*Beddington, 1983*; *Osorno et al., 2012*). In

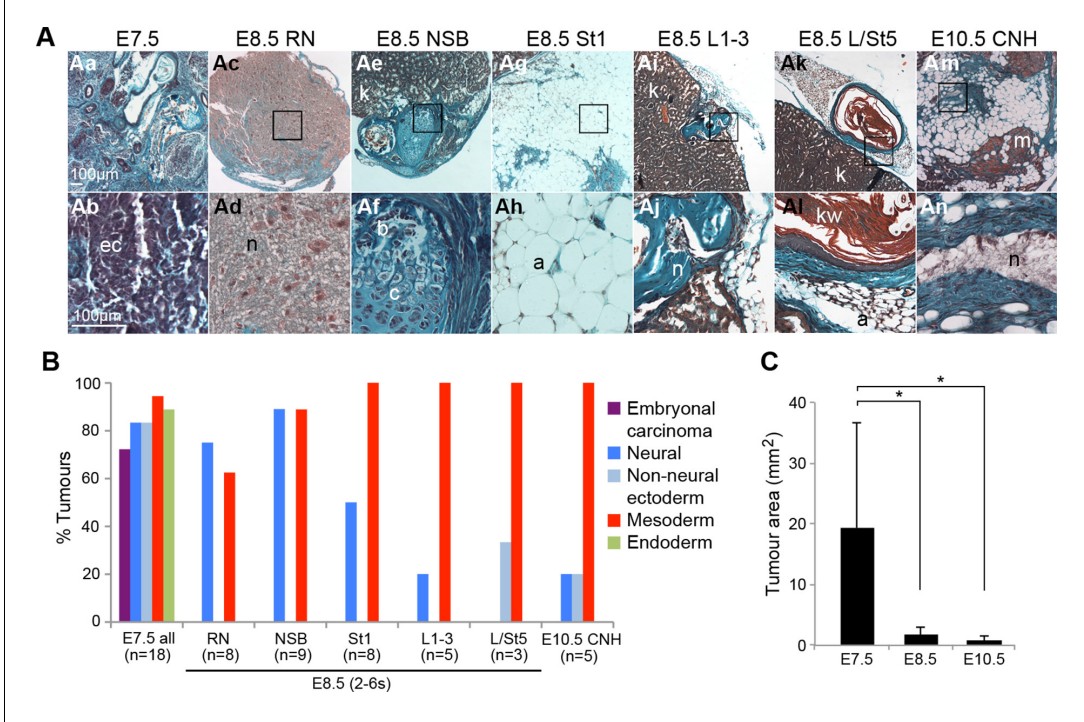

**Figure 2.** The potency of NM-fated regions is restricted to neural and mesodermal lineages. Grafts of primitive streak and tail bud regions to the kidney capsule. (**A**) Masson's trichrome-stained tumour sections derived from the indicated embryonic tissue regions. (**B**) Percentage of tumours that contain any of the scored tissues. (**C**) Average tumour area per stage. E7.5 cells give rise to larger tumours compared to tumours, derived from E8.5 or E10.5 grafts (*, p<0.01). a, adipose; b, bone; c, cartilage; ec, embryonal carcinoma cells; kw, keratin whorl; k, adult kidney; n, neural; m, skeletal muscle.

contrast, E8.5 (2–6 somite) grafts gave rise to smaller tissue masses containing only well-differentiated tissues and no EC cells. NSB, CLE (L1-3) and most (4/5) CNH grafts gave rise only to neural and mesodermal derivatives, although one CNH graft included keratinised epithelium, possibly through contamination from neighbouring specified surface ectoderm cells. Grafts of the caudal CLE and neighbouring midline (L/St5), both of which produce LVM (*Cambray and Wilson, 2007*), produced very small growths devoid of neurectoderm. Rostral PS (St1) grafts predominantly produced adipocytes, while rostral node (RN) grafts produced mainly neurectoderm. Therefore, NM-fated cells in ≥E8.5 embryonic regions are not pluripotent but restricted in potency to neural and mesoderm fates, while LVM-fated cells produce only mesoderm.

## Sox2[+]T[+] cells coincide with NMP-fated regions

We recently showed that coexpression of Sox2 and T colocalises with regions of NM fate in the mouse E8.5 caudal region (*Tsakiridis et al., 2014*). This combinatorial marker has also been reported to identify similar regions in zebrafish, chick and human (*Martin and Kimelman, 2012*; *Olivera-Martinez et al., 2012*). However, the spatiotemporal correspondence between NMP activity and Sox2/T coexpression has never been rigorously tested in any organism.

To determine whether Sox2[+]T[+] cells coincide with NMP activity, we assessed the spatiotemporal expression of Sox2 and T throughout axis elongation. Wholemount in situ hybridisation and immunofluorescence showed that Sox2[+]T[+] cells were first detected in the epiblast close to the NSB of the E7.5 (late bud stage) embryo (*Figure 3Ba,b*). At E8.5 the expression of Sox2 transcripts and protein was strong in the differentiating neurectoderm, and weaker in the CLE, while T expression in the CLE was most prominent caudally in the PS (*Figure 3A,B*). During trunk and tail formation, Sox2/T coexpression persisted in the CNH and surrounding areas (E9.5–13.5; *Figure 3B–D* and *9Ca–c*). At E13.5, just before elongation had ceased, a few Sox2[+]T[+] cells could still be detected, whereas once axial elongation was complete, Sox2[+]T[+] cells had disappeared. At E14.5, T expression was detected in the caudal notochord, while Sox2 expression was absent, consistent with the absence of the NT

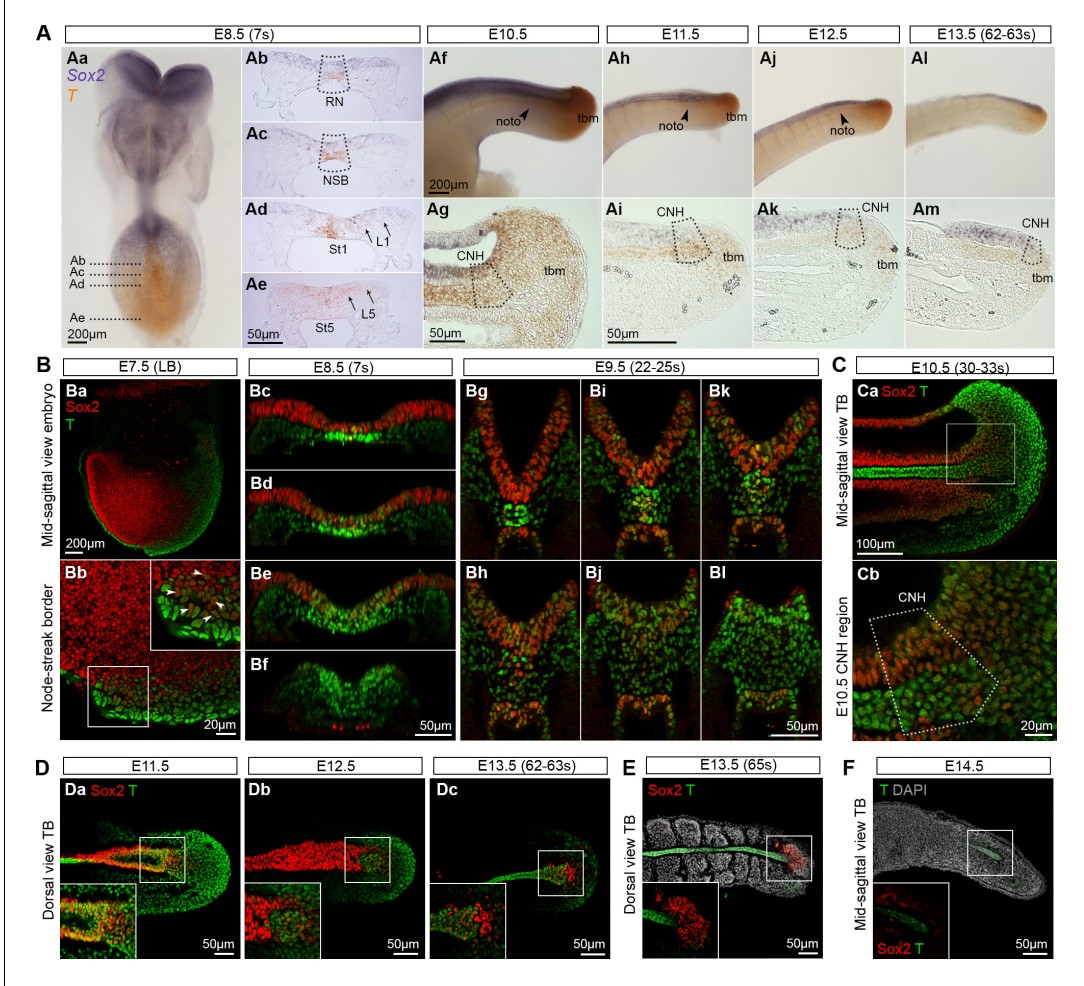

**Figure 3.** Sox2[+]T[+]cells coincide with NMP regions. (**A**) In situ hybridisation for *Sox2* and *T*. Abbreviations are the same as in *Figure 1D*. Noto, notochord; tbm, tail bud mesoderm. (**B–F**) Confocal sections of wholemount, immunostained embryos; DAPI-counterstain in grey. (**Ba**) Parasagittal section through a late bud (LB) stage embryo. (**Bb**) NSB magnified from Ba (arrowheads, Sox2[+]T[+] cells). (**Bc–l**) Transverse sections through the caudal progenitor area at E8.5 (**Bc–f**) and E9.5 (**Bg–l**) show double positive cells in the NSB and CLE. (**Ca–b**) Sagittal sections through the E10.5 tail bud. (**Da–c**) Sox2[+]T[+] cells are detected in the E11.5 tail bud, but become sparser in the E12.5 and E13.5 (up to 63s) tail bud. (**E**) Overlapping Sox2 and T expression has all but disappeared in the E13.5 (65s) tail bud in which somitogenesis is complete. (**F**) No Sox2[+]T[+] cells were detected in the E14.5 tail tip. Inset, shows the overexposed red field (Sox2).

The following figure supplement is available for figure 3:

**Figure supplement 1.** Quantifying Sox2[+]T[+]cells during axis elongation.

by this stage (*Figure 3D–F* and *Figure 3—figure supplement 1A*). Thus, Sox2[+]T[+] cells are found at all known locations and times that NMPs are present.

Use of an automated segmentation algorithm allowed us to further analyse and quantify T and Sox2 expression in confocal z-stacks throughout axis elongation (*Figure 3—figure supplement 1B* and Materials and methods). The majority of Sox2[+]T[+] cells described a 'U' shape in the epiblast at E8.5 including the NSB and L1-3, coinciding with NMP locations (*Figure 4Aa,b*). The total number of cells labelled with either marker increased up to E10.5, decreasing thereafter, while Sox2[+]T[+] cell numbers showed a similar profile but peaked at E9.5. Sox2[+]T[+] cells were rare at E13.5, and undetectable by E14.5 (*Figure 4B,C*, *Figure 3—figure supplement 1C–E* and *Video 1*). Thus the number of putative NMPs peaks during mid-trunk formation. We further quantified the levels of each transcription factor per cell during the PS-to-TB transition at E8.5–10.5. At E8.5 the levels of T were lower in the lateral than the medial CLE and the PS. T expression was high and ubiquitous in L/St5.

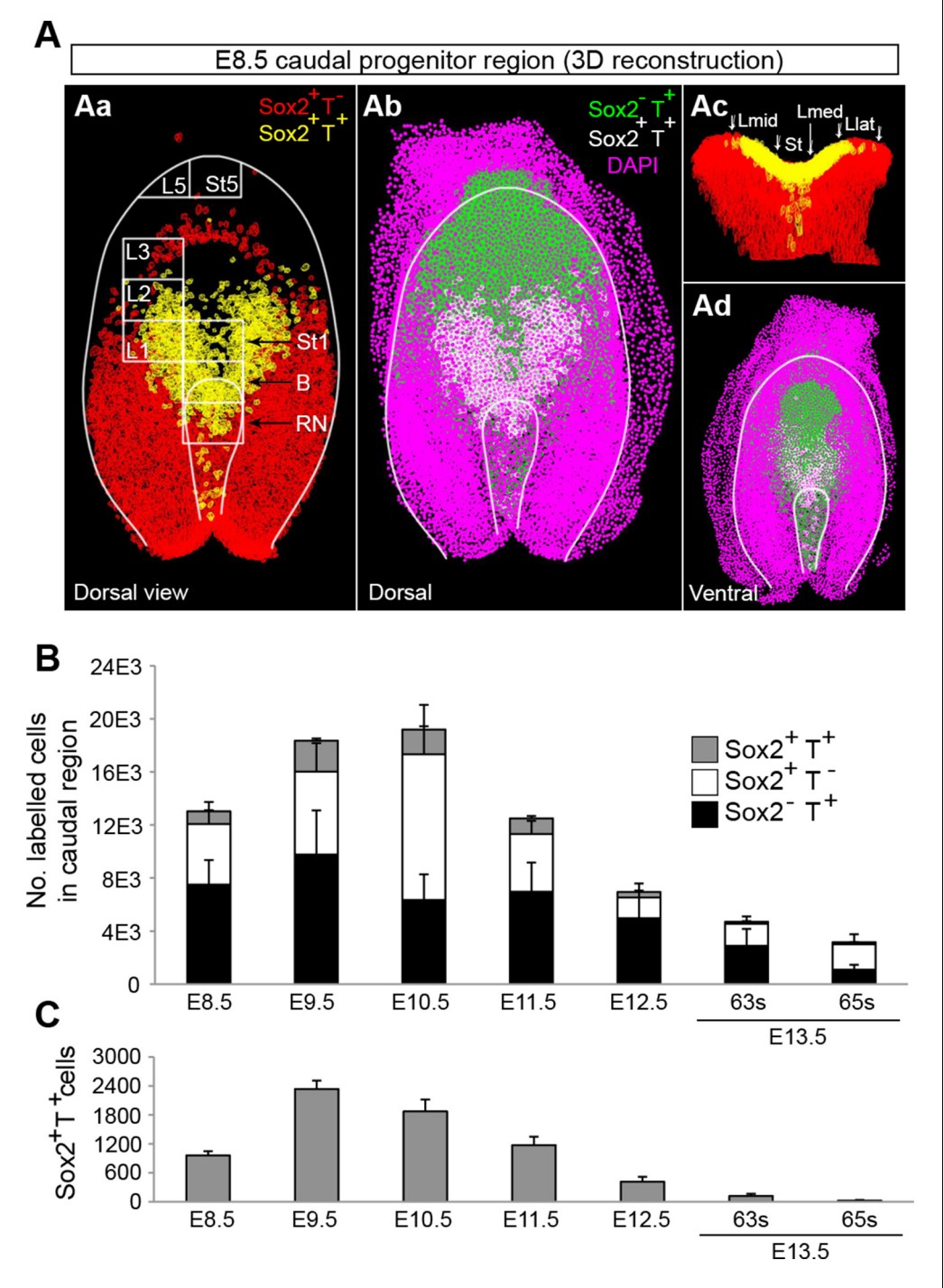

**Figure 4.** Sox2+T+ NMPs peak at mid trunk formation. (**A**) 3D reconstruction of the E8.5 caudal region. (**Aa–b**) Dorsal view. (**Ac**) Frontal view along the PS and CLE. Colours show different thresholded Sox2/T populations. (**B–C**) Quantification of different Sox2/T populations in the caudal embryo shows a peak in overall cell labelling at E10.5 (**B**). Sox2+T+ cell numbers are highest at E9.5 (**C**). Data in graphs is shown as the mean ± s.d. See also *Figure 3—figure supplement 1B–E* and *Video 1*.

The highest levels of T were found in the notochord. Sox2 was detected in a rostral-to-caudal gradient, with higher expression levels found rostrally (*Figure 5A*). Strikingly, Sox2+T+ cells were excluded from high Sox2 and high T expressing regions (*Figure 5A–C*).

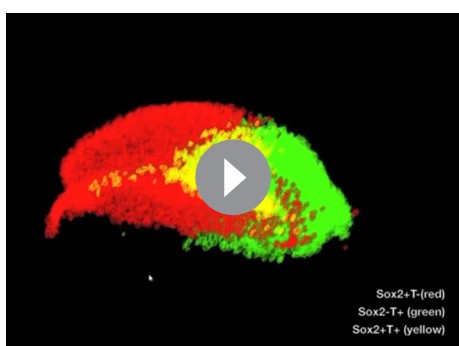

**Video 1.** Sox2 and T expression domains during axis elongation. 3D reconstruction of the caudal progenitor region at different stages of axis elongation. (00:00) E8.5 (2-5s); (00:38) E9.5; (01:02) E10.5; (01:26) E11.5; (01:47) E12.5; (02:08) E13.5 (62-63s); (02:33) E13.5 (65s).

We observed small numbers of $Sox2^+T^+$ cells in regions not expected to harbour NMPs, such as the midline PS at E8.5, suggesting that although NM progenitor identity and Sox2/T positivity substantially overlap, either this correspondence is not absolute, or previous grafting studies (*Cambray and Wilson, 2007*) may not detect low proportions of NMPs. Using a *Cited1-CreER^{T2}*-labelling system (tracing mesodermal and hindgut lineages), Garriock et al. found that some ingressed PS cells behave similarly to cells in the epiblast when *Wnt3a* is lost (*Garriock et al., 2015*), implying that some $Sox2^+T^+$ cells in the PS might retain NMP potential during early mesoderm differentiation. Interestingly, at E8.5, the numbers of $Sox2^+T^+$ cells were highest in L1, declining towards L3. Furthermore, the medial CLE contained more $Sox2^+T^+$ cells than lateral CLE ( *Figure 3Bc–f* and *4A*). Together with the graded rostral-to-caudal decline in Sox2 and increase in T, these results suggest cellular heterogeneity within the CLE.

## Fate in the caudal lateral epiblast is highly regionalised

To test whether the differences in $Sox2^+T^+$ cell numbers and relative expression within the CLE reflect differences in fate, we used homotopic grafts of groups of ~100 $GFP^+$ cells to refine the existing fate map of the CLE (donor and host locations are shown in *Figure 1A,B* and in the diagrams of *Figure 6B–D*). We analysed cell fate in the $Sox2^+T^+$ rostral CLE in both rostral-to-caudal (L1-3; *Figure 6Aa–l and 6B*) and medial-to-lateral axes (Lmed and Llat; *Figure 6Am–t and 6C*) and also in the $Sox2^-T^+$ caudal CLE and streak (L/St5; *Figure 6Au–x and 6D*). To check the accuracy of dissection, we performed immunohistochemistry on dissected L1-3 and L/St5 pieces. While $Sox2^+T^+$ cells were abundant throughout L1-3 regions (n = 7), no double positive cells were detected in L/St5 pieces (n = 5; *Figure 6—figure supplement 1*). In general, homotopic grafts incorporated well in cultured embryos (37 incorporated/42 grafts performed; *Figure 6—figure supplement 2* and *Figure 6—figure supplement 3*).

L1-3 homotopic grafts (*embryos 1.01–1.21*; $n_{sections}$ = 613) were sectioned and scored for the presence of incorporated cells. Five of 11 L1 grafts (L1-axis, 'L1A' grafts) gave rise to differentiated regions of the axis and did not contribute to the TB. These contributed extensively and unilaterally to neurectoderm (*Figure 6Aa–d*), and at lower frequency to mesoderm, predominantly (in 31/36 sections) unilaterally (*Figure 6—figure supplement 4A*). This pattern resembled that of homotopic grafts to the immediately rostral Lateral Border (LB) region (*Cambray and Wilson, 2007*). The failure of L1A grafts to contribute to the TB shows that these cells are en route for exit from the progenitor region, and the unilateral mesoderm contribution suggests that this mesoderm was already formed and carried alongside the CLE graft rather than ingressing through the PS. This is in agreement with previous reports that the presomitic mesoderm underlying the CLE contributes only to short stretches of axial tissue (estimated ~6 somites) (*Nicolas et al., 1996*; *Tam, 1986*; *1988*) (see also *Figure 6—source data 1*).

The remaining six L1 grafts (L1-axis/tail, 'L1AT'), as well as 9/10 L2-3 homotopic grafts, contributed from a variable rostral limit as far as the tail bud (*Figure 6Ae–l*). These grafts all contributed extensively and bilaterally to mesoderm, the latter indicating that the mesodermal component had passed through the PS (*Figure 6—figure supplement 4A*). Nearly all (15/16) L1AT and L2-3 grafts contributed to the TBM, but only L1AT grafts consistently colonised the CNH, indicating that L1 contains putative NMPs (*Figure 6B*). Strikingly, only L1 grafts showed significant neural contribution in the axis, while both L1 and L2-3 grafts contributed to mesoderm. Thus, most neural-fated progenitors are in the rostral-most CLE and the boundary between short-term (LB and L1A) and long-term (L1AT) progenitors lies within the L1 region.

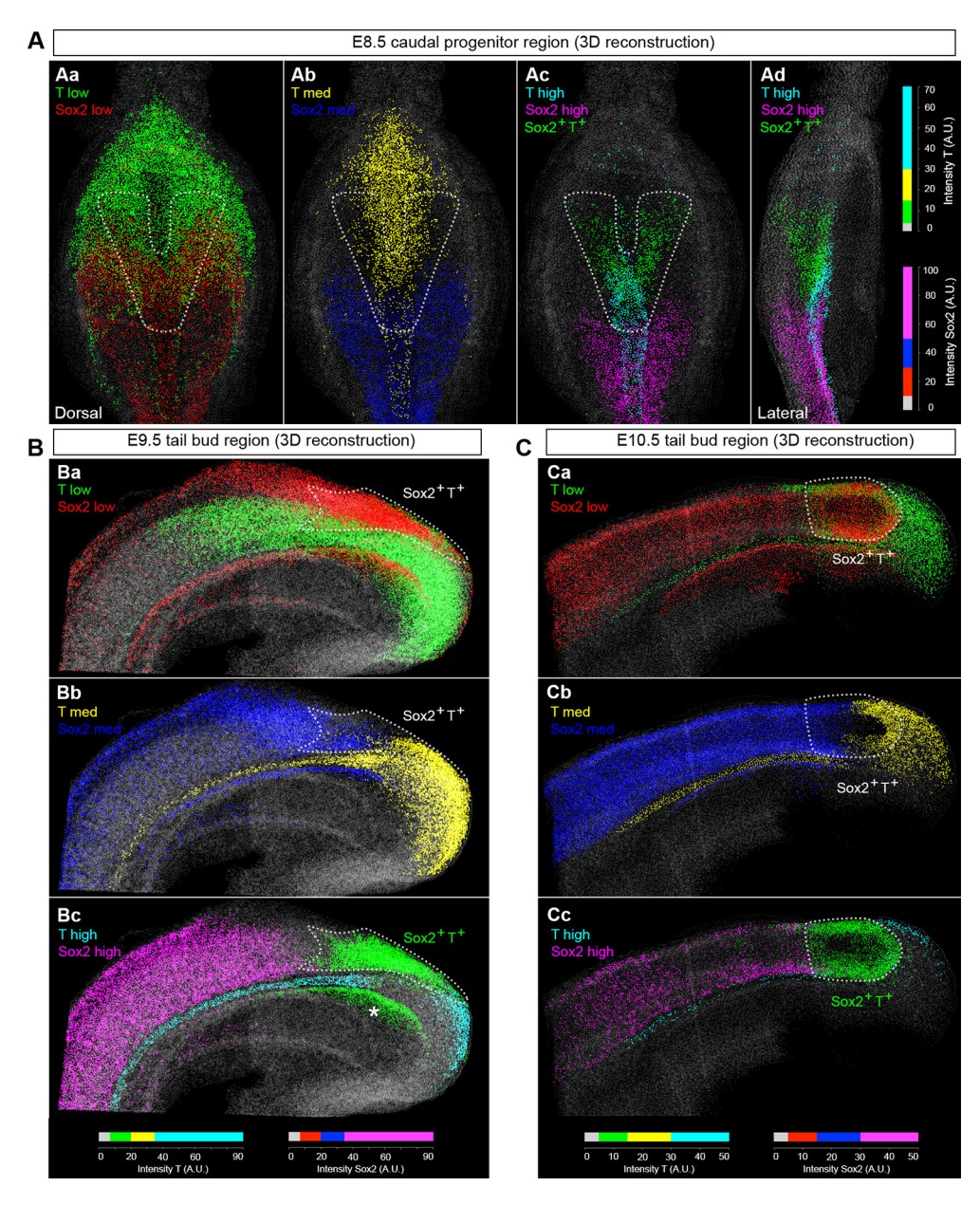

**Figure 5.** NMPs express low levels of T and Sox2. (**A–C**) 3D analysis showing the relative levels of Sox2 and T protein in the E8.5–10.5 caudal region. Sox2+T+ cells express low-to-medium levels of both transcription factors (green cells in **Ac**, **Ad**, **Bc** and **Cc** represent Sox2+T+ cells, indicated by the area within the white dotted line). Colours represent the intensity range shown as arbitrary units (AU). The lower threshold for positivity was calculated as before (see *Figure 3—figure supplement 1B*), with the maximum corresponding to the highest intensity recorded in the z-stack. Asterisk, Sox2+T+ cells found in the dorsocaudal part of the gut; grey, segmented nuclear volumes negative for either transcription factor.

Grafts to test the mediolateral fate of CLE cells (*embryos 2.01–2.08;* n<sub>sections</sub> = 196; *Figure 6Am–t, 6C*) did not show any major differences in neural or mesoderm contribution in L1/2med grafts compared to the L1-3 grafted tissue. However, when donor cells were grafted in a more lateral position (L2lat), their descendants colonised mainly the lateral and dorsal neural tube. In contrast, contribution to the mesoderm was low, largely unilateral and found at the rostral limit of the labelled part of the axis (in 10/12 sections; *Figure 6—figure supplement 4B*), suggesting this contribution might

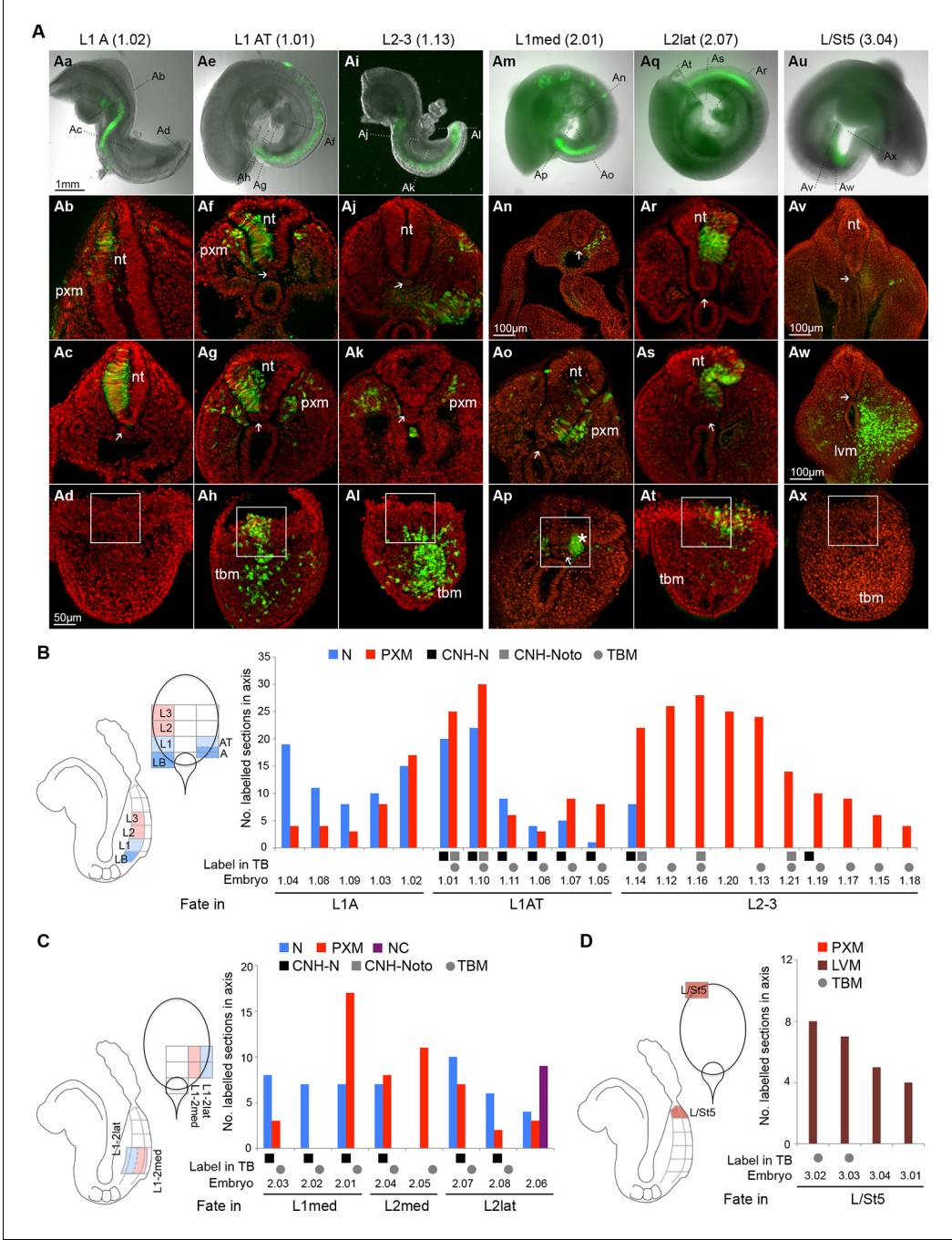

**Figure 6.** Fate in the caudal lateral epiblast. (**A**) Representative examples of homotopic CLE grafts (numbers identify individual embryos in **B–D**). Cell fate in the rostral CLE in rostral-to-caudal (**Aa–l**) and medial-to-lateral direction (**Am–t**). Cell fate in the caudal-most lateral epiblast (**Au–x**). Arrows show the notochord; white boxes the CNH. lvm, lateral and ventral mesoderm; nt, neural tube; pxm, paraxial mesoderm; tbm, tail bud mesoderm. (**B–D**) Diagrams indicate the graft type performed, the shading within them, the predominant prospective fate of each region (pink, mesoderm; light blue, neuromesodermal; dark blue, neural; light brown, lateral/ventral mesoderm). Graphs display single grafted embryos and their contribution in the differentiated axis to the neurectoderm (N) and mesoderm (PXM) or both. Numbers below the bars indicate the graft series followed by an individual embryo identifier (e.g. *embryo 1.01*). Below the x-axis, graft cell contribution in the TB (dorsal part of the CNH, ventral (notochordal) part of CNH and TBM), is represented by a black square, grey square or grey circle, respectively. (**B**) Fate in L1, L2 and L3. L1 grafts gave rise to two distinct contribution patterns, L1A (axis only) and L1AT (axis and tail bud). (**C**) Fate in either the medial or lateral half of L1 or L2. One graft (*embryo 2.06*) contributed to the neural

*Figure 6 continued on next page*

*Figure 6 continued*

crest (NC). (D) Grafts of L/St5 contributed entirely to the lateral/ventral mesoderm (LVM). Unilateral versus bilateral PXM contribution is shown in ***Figure 6—figure supplement 4***.

The following source data and figure supplements are available for figure 6:

**Source data 1.** Presomitic mesoderm contamination in CLE grafts.

**Figure supplement 1.** Rostral CLE tissue contains Sox2$^+$T$^+$ cells.

**Figure supplement 2.** Embryo numbers and section count in different grafting experiments.

**Figure supplement 3.** Homotopic and heterotopic grafts incorporate well into host embryos.

**Figure supplement 4.** Fate of the CLE progenitors in the paraxial mesoderm.

have originated largely from co-grafted committed presomitic mesoderm cells at the start of culture. In one grafted embryo (*embryo 2.06*), donor cells contributed to the neural crest (***Figure 6—figure supplement 4C***). Thus, while NM fate is found in both medial and lateral CLE, cells in lateral positions are more likely to adopt neural (particularly dorsolateral) fates.

Lastly, we assessed the fate in the caudal extreme of the CLE (L/St5; *embryos 3.01–3.04*; n$_{sections}$ = 103; ***Figure 6Au–x,D***). All homotopic L/St5 grafts displayed consistent LVM contribution to a small stretch of the post-hindlimb axis. While homotopic L2-3 grafts displayed low but reproducible LVM contribution distributed throughout the labelled stretch of axis (20% of sections in 80% of embryos; ***Figure 6Aj*** and ***Figure 10—figure supplement 1A***), LVM contribution was very rare in L1 grafts (<1% of sections in 9% of embryos; ***Figure 10—figure supplement 1A***). Thus, L/St5 is the only region to contribute exclusively to LVM (***Figure 6Au–w***, ***Figure 10—figure supplement 1A*** and ***Cambray and Wilson (2007)***).

These homotopic grafts show that neural and mesodermal fate is more highly regionalised than previously appreciated. In the rostral CLE (L1-3), cells in the caudal and medial parts of the CLE are more likely to adopt paraxial mesodermal fates, while the rostral and lateral CLE favours neural fates. These fates correlate with the relative levels of Sox2 and T, higher levels of which appear to predict neural/ neuromesodermal and mesodermal fates respectively. The caudal CLE (L/St5), which expresses no Sox2, adopts exclusively lateral/ventral mesoderm fates.

## Cells in the rostral half of the CLE have equivalent NM differentiation potential

The regional diversity of fates in the CLE prompted us to test whether CLE cells were inherently biased towards neural or mesodermal lineages. Since the NSB contributes robustly to both neural and mesodermal lineages (***Cambray and Wilson, 2007***) we used this site to test the plasticity of CLE cells by performing CLE-to-NSB grafts. As for homotopic grafts, heterotopic grafts incorporated well in the axis (57 incorporated/61 grafts performed). Strikingly, the relative contribution to neural or mesodermal fates in the axis varied extensively between grafts of rostral CLE (L1-3) to NSB (*embryos 4.01–4.09*; ***Figure 7A–B***).

Separating L1 from L2-3 (*embryos 5.01–5.18*) tested whether this observed variation was influenced by the origin of the grafted tissue. However, both L1-to-NSB and L2-3-to-NSB grafts appeared indistinguishable, with about half of the embryos in each class containing neural contribution of differing extents. We further tested whether an intrinsic mediolateral bias could instead explain the variation seen above. Medial or lateral L1 or L2 pieces were grafted to the NSB. However, these grafts did not show consistent differences in neural or mesodermal differentiation, and some grafts reversed their predicted fate (***Figure 7C–D***). Thus neither the neural fate of L1 and Llat, nor the mesodermal fate of L2-3 were retained after heterotopic grafting to the NSB, and therefore could not account for the variable neural and mesodermal contribution seen in individual grafted embryos. Consequently, the fate differences seen in different CLE areas cannot be explained by ≥2 localised populations of restricted potential.

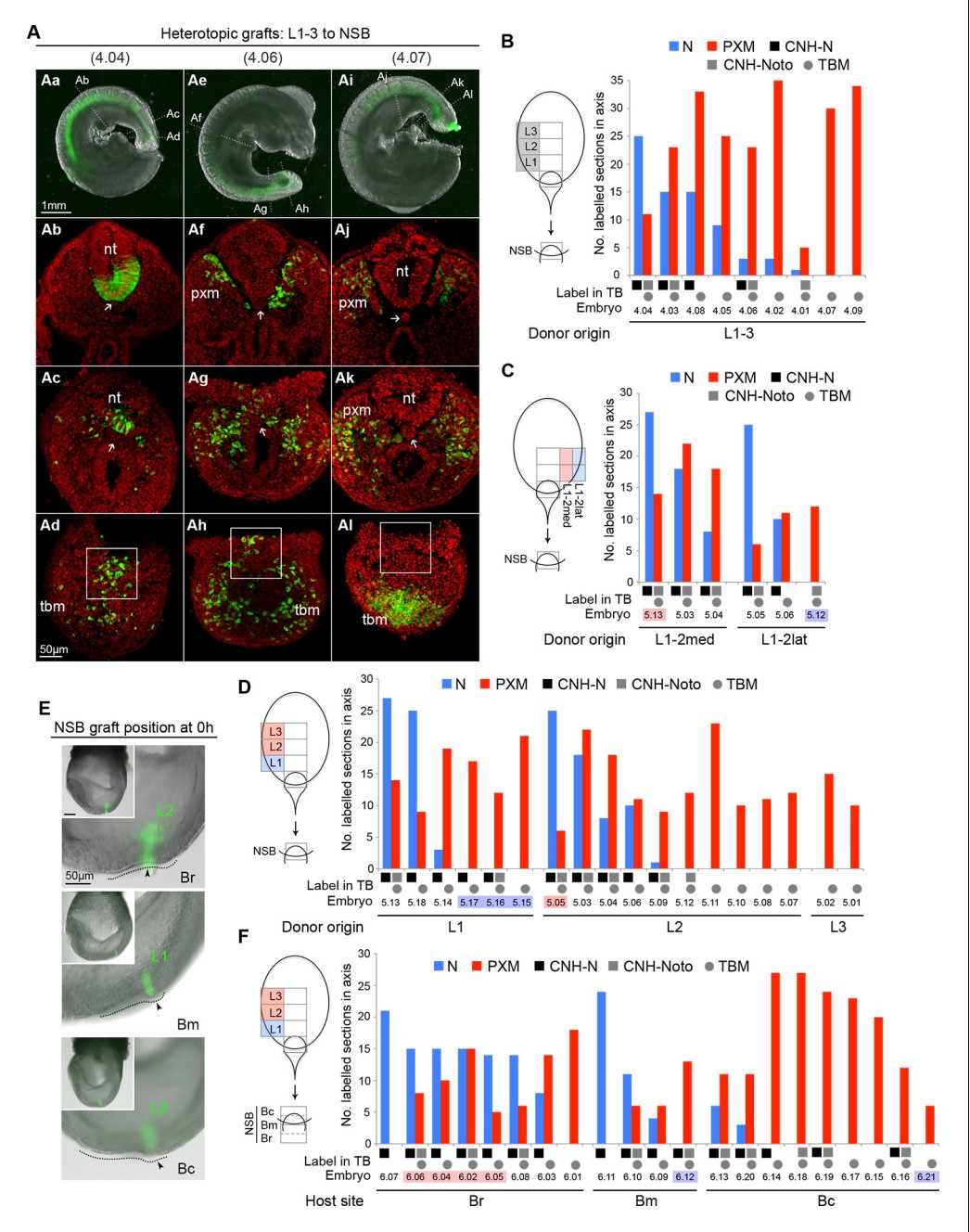

**Figure 7.** Plasticity of L1-3 cells on heterotopic grafting to the NSB. (**A**) Representative examples of heterotopic CLE grafts (numbers identify individual embryos in **B**). Three different contribution patterns were observed: mainly neural (**Aa-d**), neuromesodermal (**Ae–h**) and mainly mesodermal (**Ai–l**). (**B–D** and **F**) Scoring of GFP⁺ cell contribution in CLE-to-NSB heterotopic grafts. Graph format is the same as in *Figure 6*. Highlighted numbers indicate embryos in which the graft fate has changed from that predicted by the fate map upon heterotopic grafting. (**B**) Grafts of L1, 2 or 3 to the NSB. (**C**) Grafts of medial or lateral halves of L1 or L2 to the NSB. Note that these embryos are also scored with respect to their rostrocaudal origin in **D**. (**D**) Grafts of L1, 2 or 3 to the NSB. Note that some of these embryos are also scored according to their mediolateral origin. For example *embryo 5.13* received a graft of cells from medial L1, and showed high neural contribution and is therefore highlighted as 'changed fate' in **C** but not in **D**. (**E**) Graft position at the start of culture (to Br, Bm or Bc). Donor origin is shown in green. The crown (arrowhead) of the node (dotted line) was used as a landmark for placing GFP⁺ donor cells (green). (**F**) Grafts of L1, 2 or 3 to different aspects of the NSB shows the fate of rostral CLE cells is dependent on the environmental cues of the host.

## Neural versus mesodermal outcome is influenced by NSB position

The NSB forms the junction between the streak, containing mesoderm-fated cells, and the caudal part of the node, which, according to fate maps in chick (*Selleck and Stern, 1991*) contains neural-fated cells. In chick, the exact location in the node influences cell fate. To test whether exact position within the NSB could predict neural versus mesodermal contribution, we grafted L1-3 cells to the rostral, middle, or caudal aspects of the NSB (Br, Bm or Bc respectively; *Figure 7E*. See also *Figure 1Bb*). In contrast to the previous series, these grafts produced distinct and reproducible patterns of labelled cell distribution (*embryos 6.01–6.21*). Rostral CLE cells placed in either Br or Bm tended to contribute entirely, or predominantly, to neural tube, while grafts to Bc resulted in a high proportion of embryos with mesoderm-only contribution. In several embryos, the position within the NSB reversed the predicted fate bias of L1 towards neural and L2-3 towards mesodermal fates: 5/5 L2-3-to-Br grafts contributed to the NT, while *embryo 6.21*, an L1 graft, gave rise exclusively to mesoderm (*Figure 7F*). Thus our data indicate that assignment of rostral CLE cells to neural or meso-dermal fates is dependent on environmental cues, with no detectable cell-intrinsic biases. This data therefore indicates a single NMP type capable of context-specific neural or mesodermal differentia-tion and retention in the progenitor region.

Homotopic NSB grafts give rise to a short stretch of notochord (*Cambray and Wilson, 2007*) but it was not clear from these grafts whether this was derived from the dorsal or ventral layer of the node. CLE-derived cells were detected in the caudal notochord (*Figure 6—figure supplement 3*). However, these cells did not express the notochord marker Foxa2, nor the high levels of T expressed in neighbouring host notochord cells. Therefore, although they were able to enter the notochord domain, they were unable to differentiate correctly, suggesting that E8.5 NMPs are not notochord progenitors. This is consistent with fate mapping studies indicating that the posterior notochord is derived from nodal positive cells in the mesoderm adjacent to the node (*Brennan et al., 2002*) and that convergent extension is the main driver of notochord extension at early somite stages (*Yamanaka et al., 2007*). Together, these results suggest that from E8.5 onwards the notochord may elongate primarily as a result of rearrangement of pre-existing notochord progenitors rather than *de novo* addition of cells from the epiblast layer.

## Dorsolateral bias in NT and CNH is reset upon transplantation to the midline NSB

To further test the context specificity of NMPs, we compared the overall neural versus mesoderm and CNH contribution of all graft series. Grafting L2-3 progenitors to the midline NSB not only changed their respective neural versus mesodermal contribution (shown as the percentage of all sec-tions for a given series; *Figure 8A*), but also increased their CNH contribution (from 20% to 42% of sections), suggesting that a midline position favours NMP maintenance throughout axis elongation.

Further examination of homotopic CLE grafts indicated that they contributed to more dorsolat-eral positions than grafts to the NSB. To quantify this observation, we measured the dorsoventral extent of GFP$^+$ cell contribution in all scoreable sections of CLE homotopic and CLE-to-NSB hetero-topic grafts (*Figure 8B–C*). The frequency of GFP$^+$ cell colonisation of any NT position from 0% (dor-sal righthand extreme) through 50% (ventral NT) to 100% (dorsal lefthand extreme) showed graft-type specific differences. CLE homotopic grafts predominantly contributed to lateral regions of the NT in the axis, but shifted towards the midline in the tail bud (*Figure 6Aa–h and Aq–t*), indicating that there is a net displacement of at least some of the CLE cells towards the midline as axis elonga-tion proceeds, perhaps to replace cells that have exited to the mesoderm via the midline PS. CLE cells grafted to the NSB appeared to have erased their positional bias, as NT contribution was pre-dominantly midline in both axis and TB sections (*Figure 8C*); a contribution pattern which strongly resembled NSB homotopic grafts previously described in (*Cambray and Wilson, 2007*). Thus, as positional information in neural progenitors can be reset, it argues for the existence of a single NMP identity whose fate and dorsoventral NT position is instructed by positional cues.

## Wnt/β-catenin signalling is required for mesodermal fate in NMPs

To define the molecular basis of NMP fate choice, we examined the role of Wnt/β-catenin signalling. Inhibiting Wnt signalling can divert predominantly mesoderm-fated cells in early zebrafish embryos towards neural fates (*Martin and Kimelman, 2012*). Conditional deletion of Wnt3a or β-catenin in

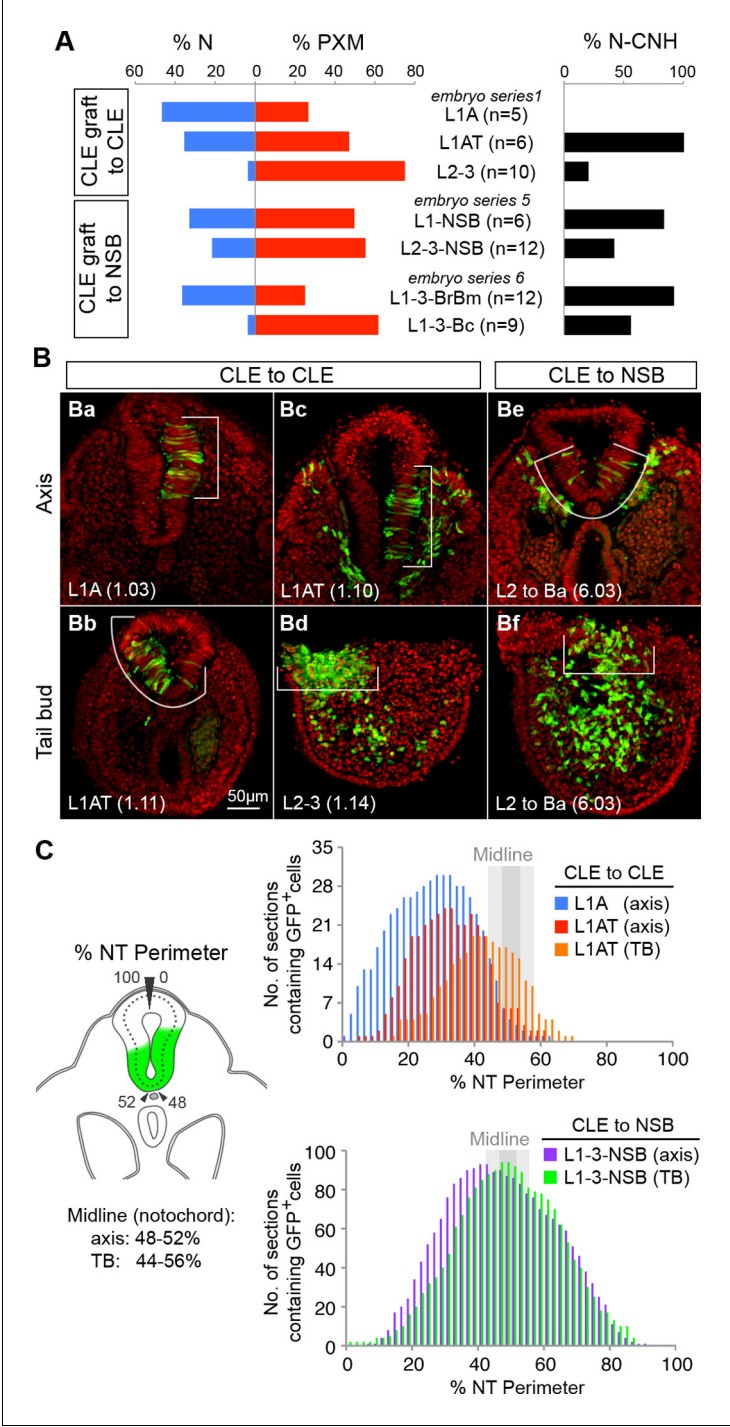

**Figure 8.** Dorsolateral bias of L1-3 cells is reset upon grafting to the midline. (**A**) Quantitative analysis of L1-3 graft contribution. Left, the percentage of neural (N) vs paraxial mesoderm (PXM) contribution in all scorable sections. Right, the percentage of embryos with dorsal (neural) CNH contribution. Labels show the embryo series and the graft performed (with n, the number of embryos). (**B**) Representative examples of dorsolateral NT distribution of grafted cells to either one lateral side (CLE to CLE) or to the midline (CLE to NSB). White lines show the extent of NT contribution. (**C**) Left, diagram shows the method used to score mediolateral NT contribution of donor cells (green), expressed as a percentage of the NT perimeter. Right, graphs display the number of sections containing GFP$^+$ cells at defined positions along the NT. Colours represent graft type (blue, L1A homotopic grafts; red and orange, L1AT homotopic grafts; purple and green, L1-3 to NSB heterotopic grafts). Sections in the TB are

*Figure 8 continued on next page*

*Figure 8 continued*

represented separately from those anterior to the TB (termed 'axis'). The average notochord position at the ventral midline is shown by dark grey (axis) and light grey (TB) shading.

the T-expressing population, followed by lineage tracing, has confirmed that, in mouse, β-catenin is required for mesoderm differentiation in PS/TB cells (*Garriock et al., 2015*). However the above study does not directly address the role of β-catenin in NMPs, which form a subset of the $T^+$ population. Therefore we investigated the role of Wnt/β-catenin specifically in mouse NMPs. We utilised embryos carrying a conditional (floxed) *β-catenin* mutation, a conditionally active (floxed stop) *GFP* marker and a ubiquitously expressed tamoxifen-inducible *Cre* transgene (termed here *βcatCKO: sGFP*), which delete β-catenin and activate *silent GFP (sGFP)* upon 4-hydroxytamoxifen (4-OHT) treatment (*Figure 9—figure supplement 1A–C*). We dissected either *βcatCKO:sGFP* or constitutively active GFP transgenic control (*AGFP7*) L1-3 pieces and grafted them to the rostral streak (St1-3). Embryos grafted with *βcatCKO:sGFP* cells were treated with 4-OHT for the first 8 hr of a 48-hr culture period, which proved sufficient to delete β-catenin in the majority of cells (*Figure 9—figure supplement 1D*). For an overview of the experimental set-up and the number of grafts performed, see *Figure 9—figure supplement 2A–E*.

As expected, all control L1-3 to St1-3 *AGFP7* grafts (n = 4) contributed extensively to the PXM and TBM. With the exception of a few cells in a single embryo, the CNH was not colonised (*Figure 9—figure supplement 2F*). Similarly, the 4-OHT-treated *βcatCKO:sGFP* donor cells that exited the PS early, and thus contributed to rostral parts of the axis, produced exclusively mesoderm. In contrast, those exiting later to caudal regions produced only neurectoderm ($n_{embryos}$=11; 9/9 incorporated grafts switched from PXM to neural fate, *Figure 9A*). $GFP^+$ cells generally showed undetectable levels of β-catenin, but expressed the cell adhesion molecule N-cadherin at levels similar to neighbouring wildtype cells, suggesting that their cell adhesion properties remain intact (*Figure 9B*). Graft-derived cells in the PXM correctly expressed Pax3, while those in the NT expressed neural markers (*Figure 9B* and *Figure 9—figure supplement 2G*), showing that β-catenin deletion does not preclude either neural or mesoderm differentiation after cells have exited from the NMP compartment. Thus, the decision of NMPs to form mesoderm depends absolutely on β-catenin, and in its absence, cells differentiate to neural derivatives.

To determine the effect of β-catenin on $Sox2^+T^+$ putative NMPs we deleted β-catenin in whole ex-vivo cultured *βcatCKO* embryos between E8.5–9.5 and quantified Sox2 and T expression as before. Compared to wildtype controls, the notochord extended further caudally than in wildtype embryos and the size of the NMP region was reduced in 4-OHT-treated *βcatCKO* and $Sox2^+T^+$ cells appeared scattered throughout the CLE (*Figure 9C* and *Figure 9—figure supplement 3B*). We observed a significant drop in $Sox2^+T^+$ cell numbers in E9.5 treated samples (*Figure 9C* and *Video 2*). Moreover, the number of $Sox2^+T^+$ cells in E9.5 β-catenin-deleted embryos was not significantly different from untreated E8.5 samples, indicating a failure to expand NMP numbers (*Figure 9D*). We also analysed the Sox2 and T single-positive populations to determine whether either was disproportionately affected. We observed a significant drop in $Sox2^-T^+$ cell numbers (*Figure 9E–F* and *Figure 3—figure supplement 1F–H*), consistent with the observation that T is a direct transcriptional target of Wnt signalling via β-catenin (*Yamaguchi et al., 1999*).

To determine the relative effects of β-catenin depletion on the different progenitor populations in the PS, we compared the different levels of T expression in WT and 4-OHT-treated *βcatCKO* embryos. T immunostaining intensity per cell was subdivided into five levels, quantitated and plotted on the reconstructed 3D scaffold (*Figure 9G* and *Figure 9—figure supplement 3A–B*). The number of cells expressing all but the highest levels of T was significantly reduced in 4-OHT-treated *βcatCKO* embryos (*Figure 9H* and *Figure 9—figure supplement 3C*), whereas the $T^-$ fraction was significantly increased. Moreover, levels of T protein were dramatically downregulated in the area containing *βcatCKO* NMPs (*Figure 9Ga"",Ga"',Ga''',Gc,Gc"" and Gc"'*), as well as in their descendants in the PSM (*Figure 9Ga,Ga',Ga'',Gc,Gc' and Gc"*). Interestingly, the LPMP-derived ventral mesoderm of the cloaca of β-catenin-depleted embryos still retained higher levels of T compared to the PSM (compare *Figure 9Ga"" and Gc""*). Thus, Wnt signalling is required for NMP expansion, at least in part through the maintenance of T. Moreover, our data suggests the maintenance of T in

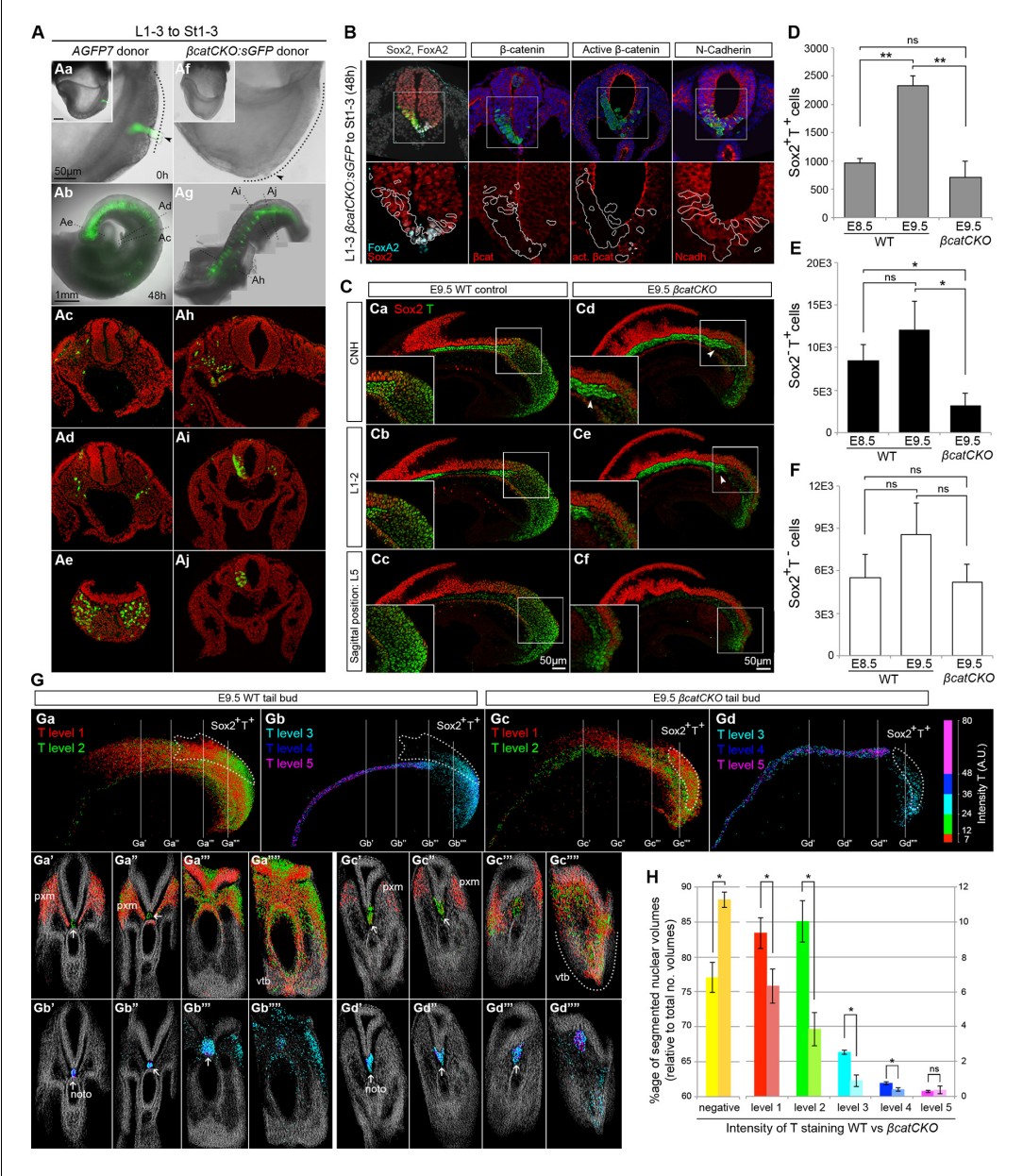

**Figure 9.** Canonical Wnt signalling mediates neuromesodermal fate decisions. (**A**) Representative examples of L1-3 to St1-3 grafts. (**Aa** and **Af**) Cells immediately after grafting. *βcatCKO:sGFP* grafted embryos were grown for 8 hr in the presence of 5μM 4-OHT after which the medium was replaced. Note grafted cells are initially GFP- and turned green upon CreER^T2 activation and LoxP recombination. Dotted line, primitive streak. Arrowhead, grafted cell position. (**Ab** and **Ag**) Whole embryo after 48 hr ex vivo culture. (**Ac–e** and **Ah–i**) Rostral to caudal transverse sections. (**Aa–e**) L1-3 *AGFP7* to St1-3 control grafts contribute only to the paraxial mesoderm, although one embryo showed minor contribution to the CNH and resembled some of the L1-3 to Bc grafts (compare *Figure 9—figure supplement 2F* and *Figure 7F*). (**Ai–Aj**) L1-3 *βcatCKO:sGFP* to St1-3 grafts shows NMPs switch from mesoderm to neural fate in the absence of β-catenin. (**B**) Marker and β-catenin expression in graft-derived *βcatCKO:sGFP* cells. Sections correspond to a similar rostral level as shown in **Aj**. Upper row, DAPI-counterstained images (grey/blue). Squares, region magnified in lower row. Lower row, immunofluorescence channel with donor cell position outlined in white. See *Figure 9—figure supplement 2* for an experimental overview, the number of embryo grafts performed and additional immunostaining results. (**C**) Immunofluorescence of T (green) and Sox2 (red) in E9.5 WT (n = 5, Ca-c) and *βcatCKO* embryos (n = 5, Cd-f) after 24 hr in vitro growth in the presence of 5μM 4-OHT. Arrowheads, notochord in treated embryos. (**D–F**) Quantitation of labelled populations in *βcatCKO* and WT embryos (*, p-value<0.05; **, p-value<0.0001; ns, not significant). See also *Figure 3—figure supplement 1* and *Video 2*. (**G**) 3D analysis showing the relative levels of T protein in E9.5 WT (**Ga** and **Gb**) and *βcatCKO* tail buds (**Gc** and **Gd**) shown as arbitrary units (AU) of staining intensity). Arrows, notochord (noto) position; grey, T⁻ cells; pxm, paraxial mesoderm; white dotted line, NMP area. (**H**) Quantification of T levels in E9.5 WT (n = 2) and *βcatCKO* tail buds (n = 4). Left bars, WT; right bars, *βcatCKO* embryos. (*, p-value<0.05; ns, not significant). See also *Figure 9—figure supplement 3C*.

*Figure 9 continued on next page*

*Figure 9 continued*

The following figure supplements are available for figure 9:

**Figure supplement 1.** Obtaining conditional β-catenin knock out embryos.

**Figure supplement 2.** Grafts using β-catenin knock out donor tissue.

**Figure supplement 3.** Levels of T in wildtype and 4-OHT-treated *βcatCKO* embryos.

LPMPs and their descendants is less dependent on β-catenin. Therefore, LPMPs may represent an alternative, β-catenin-independent route towards mesoderm formation.

## The caudal-most CLE exhibits mesoderm-restricted plasticity

The plasticity exhibited by rostral CLE cells led us to examine whether the caudal extreme of the CLE (L/St5) is similarly environment-sensitive. We grafted L/St5 cells to the rostral PS (St1-3; *embryos 7.01–7.03*); a region that contributes extensively to paraxial mesoderm (*Cambray and Wilson, 2007*). In this location, L/St5 cells contributed robustly to both PXM and LVM. Similar to homotopic St1 grafts, L/St5 to St1-3 grafts contributed to the TBM but not CNH (*Figure 10Aa–e* and *Cambray and Wilson (2007)*). However contribution to the TBM was limited (*Figure 10Ae,B,D* and *Figure 10—figure supplement 1B*). We did not observe obvious differences in the mediolateral distribution of PXM descendants in any of the L/St5 grafts (*Figure 10—figure supplement 2*). Hence, these caudal-most CLE progenitors show plasticity within the mesoderm lineage.

To test whether these cells depend on active β-catenin to prevent neural differentiation, 4-OHT-treated *βcatCKO:sGFP* L/St5 cells were grafted more rostrally in the PS. Unlike rostral CLE, β-catenin deleted L/St5 cells formed only PXM and LVM without NT differentiation (*Figure 10Af–j*). Thus, β-catenin is not required for L/St5 cells to undergo mesoderm differentiation. To determine whether L/St5 progenitors are capable of forming neural tissue using regional cues similarly to the rostral CLE, we grafted L/St5 cells to the rostral border (*embryos 8.01–8.09*). All L/St5 to Br grafts showed exclusively mesodermal contribution except for a small patch in the NT of one embryo (*embryo 8.03*; *Figure 10Ak–o,C–E* and *Figure 10—figure supplement 1C*). Thus, environmental cues can switch the exclusive LVM fate of at least some L/St5 cells to include PXM, but they do not adopt neural fates, even where L1-3 cells undergo predominantly neural differentiation. We therefore term this distinctly-committed population lateral/paraxial mesoderm progenitors (LPMPs).

We have shown above that Sox2 is absent in the caudal-most tip of the CLE (*Figure 3Bf* and *4A*). This transcription factor, together with Sox3, is required for neural differentiation, and can reinstate neural differentiation when ectopically expressed in emergent PXM (*Takemoto et al., 2011*; *Yoshida et al., 2014*). To test whether Sox2 is sufficient to confer neural fate on L/St5 cells, dissected L/St5 regions were electroporated with a CAG-Sox2-T2A-tdTomato expression plasmid and grafted to the rostral aspect of the border, which favours neural fate in grafted L1-3 cells. Cells that did not take up the vector incorporated well and contributed to the PXM as before. In contrast, tdTomato-expressing cells formed non-integrated clumps along the axis (*Figure 10F*). No evidence of neural differentiation was apparent in electroporated cells (n$_{embryos}$ = 6). Thus, ectopic expression of *Sox2* is unable to override the inability of L/St5 cells to contribute to the NT.

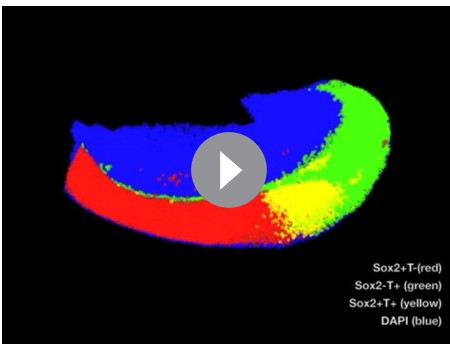

**Video 2.** The number of T$^+$ cells is affected upon β-catenin removal. 3D reconstruction of the caudal progenitor region. (00:00) E9.5 WT sample and (00:28) example of a *βcatCKO* embryo at E9.5, after it was cultured in vitro for 24 hr in the presence of 5μM 4-OHT.

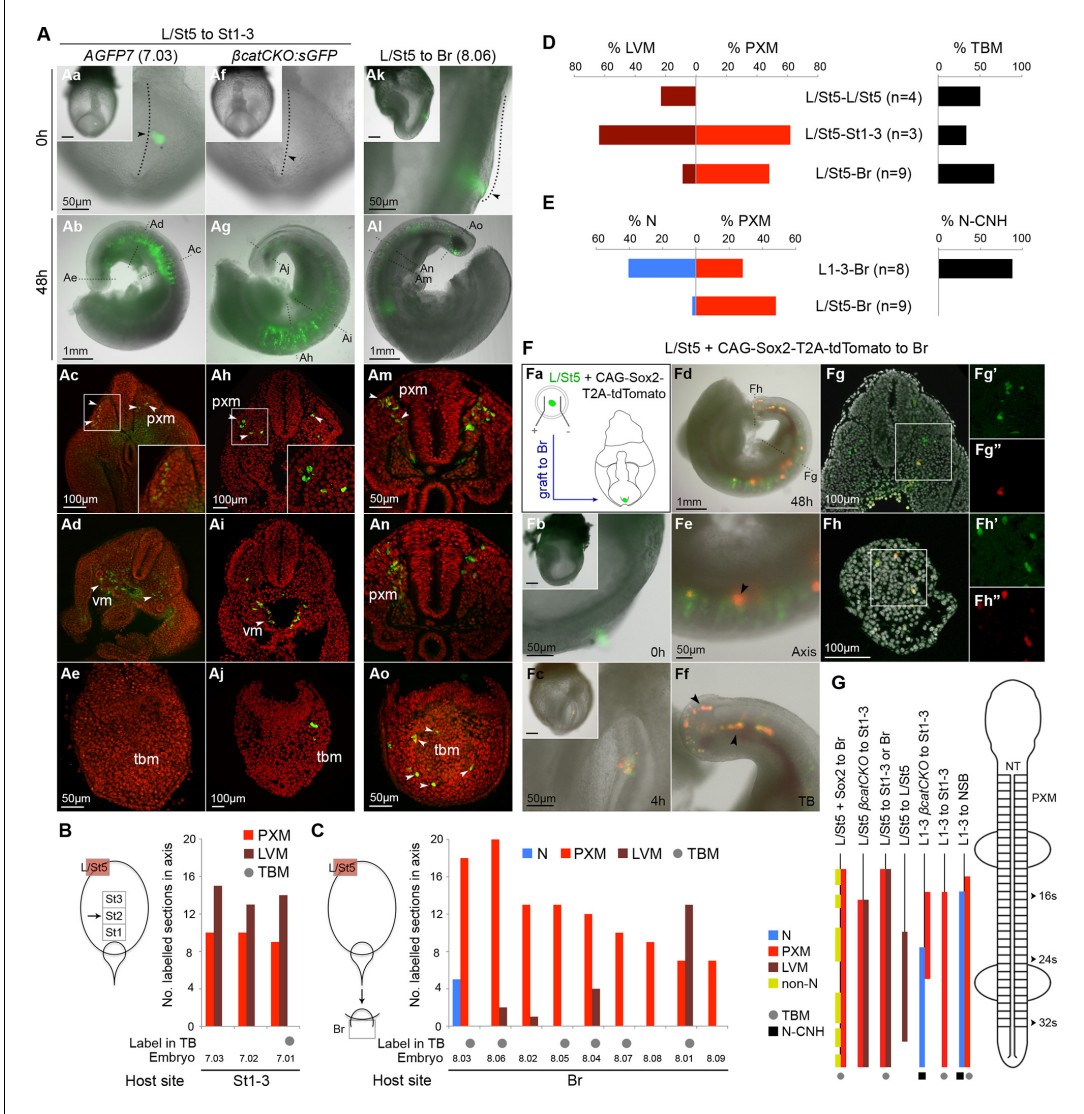

**Figure 10.** The caudal tip of the CLE shows paraxial mesoderm but not neural potency. (**A**) Representative examples of L/St5 heterotopic grafts (numbers in brackets identify individual embryos depicted in (**B–C**). (**Aa, Af** and **Ak**) Cells immediately after grafting. Arrowheads show grafted cell position. (**Ab, Ag** and **Al**) Whole embryo after 48 hr ex vivo culture. (**Ac–e, Ah–i** and **Am–o**) Rostral to caudal transverse sections. (**Aa–e**) L/St5 *AGFP7* to St1-3 grafts shows LVM-fated cells can switch fate to form PXM. (**Af–j**) L/St5 *βcatCKO:sGFP* to St1-3 grafts show a similar contribution pattern to control grafts. Non-integrated clumps near the notochord were observed in 3/7 embryos (see *Figure 9—figure supplement 2H*). (**Ak–o**) L/St5 to Br grafts show robust contribution to the paraxial mesoderm. (**B–C**) Grafting of L/St5 donor cells to the primitive streak (**B**) or to Br (**C**). Graph format is the same as in *Figure 6*. (**D–E**) Quantitative analysis of L/St5 graft contribution. Left, the percentage of PXM vs LVM (in **D**) or NT (in **E**) contribution in all scorable sections. Right, the percentage of embryos with TBM (in **D**) or dorsal CNH (in **E**) contribution. (**F**) Ectopic over-expression of Sox2 in L/St5 cells (n = 6 embryos). (**Fa**) L/St5 cells of *AGFP7* embryos were electroporated with CAG-Sox2-T2A-tdTomato plasmid before grafting to Br of WT hosts. (**Fb–c**) Electroporated cells immediately after grafting (**Fb**) and after 4 hr culture (**Fc**). (**Fd–f**) Grafted embryo after 48 hr in vitro growth. Arrowheads show areas containing not well-integrated cells (orange). Green cells were not electroporated at the start and contributed to the PXM and LVM as before. (**Fg–h**) Axis and TB sections show orange cells never contribute to the NT. (**G**) Summary of L1-3 and L/St5 heterotopic grafts. Coloured bars represent the contribution to different axial tissues. Contribution in the neural CNH and TBM is represented by a black square or a grey circle, respectively. N, neural; PXM, paraxial mesoderm; LVM, lateral/ventral mesoderm; non-N, non-neural clumps; TBM, tail bud mesoderm; N-CNH, dorsal (neural) part of the CNH.

The following figure supplements are available for figure 10:

**Figure supplement 1.** Overall lateral/ventral and tail bud mesoderm contribution.

**Figure supplement 2.** Somite contribution in heterotopic LPMP grafts.

## Discussion

Fate and potency mapping, together with genetic manipulation, have allowed us to define two populations of progenitors, NMPs and LPMPs, residing in the primitive streak region. Distinct regulatory logics govern their intrinsic abilities to respond to their environment (*Figure 11*). These lineage-restricted progenitors in E8.5 embryos emerge from the pluripotent epiblast at the beginning of somitogenesis. The lineage restriction of these populations, together with their lack of neoplastic potential, may render them useful for clinical applications.

### The Sox2$^+$T$^+$ phenotype overlaps extensively with NM potency

Despite previous reports that Sox2$^+$T$^+$ positivity identifies NMPs, no study has previously comprehensively correlated this phenotype with the known spatiotemporal characteristics of NMPs. Our immunohistochemical analysis shows that Sox2$^+$T$^+$cells are found in all NMP-containing regions: the NSB, rostral CLE, and the ectoderm of the CNH during throughout axis elongation, and are undetectable once somitogenesis ceases, suggesting they are associated with NMP activity. Therefore, are Sox2$^+$T$^+$cells equivalent to NMPs? Sox2$^+$T$^+$cells are found at low frequency in regions not expected to be NM potent: the E8.5 midline primitive streak, the region caudal to the CNH at E10.5, the hindgut and notochord (*Cambray and Wilson, 2007*; *McGrew et al., 2008*). Conversely, the rostro-lateral CLE (L1-2lat) is NM potent but contains very few Sox2$^+$T$^+$cells. Importantly, however, *T* transcripts extend more laterally in the epiblast than protein (*Figure 3A–B*; *Wilson et al. (1995)*), suggesting that Llat may already be poised to accumulate T protein. We demonstrate that the levels of T and Sox2 correlate precisely with likelihood of mesodermal and neural fate respectively, while NMPs are excluded from high-expressing areas (*Figure 5*). Moreover, the level of T in NMPs is significantly reduced when the capacity for mesodermal differentiation is lost in *βcatCKO* cells (*Figure 9G*). Finally, Sox2$^+$T$^+$ cells first appear at late neural plate stage, several hours before pluripotency is lost in the epiblast (*Osorno et al., 2012*). Thus, the Sox2$^+$T$^+$ phenotype seems to overlap extensively with (although may not absolutely define) NMPs.

Two further observations support the NMP identity of Sox2$^+$T$^+$cells. Firstly, the efficient and extensive incorporation of grafted cells, and the ability of these regions to robustly adopt the fate of their new environment in heterotopic grafts, argue that most grafted NSB and L1-3 cells are NMPs. Secondly, clonal analysis indicates that the number of NMPs present at E9.5 is approximately 2.8-fold greater than that at E8.5 (5 NM clones with a rostral limit of s10-20, versus 14 with a rostral limit of s21-30 (*Tzouanacou et al., 2009*)). This increase is similar to the 2.4-fold increase in number of Sox2$^+$T$^+$ cells during this period (960 ± 85 versus 2338 ± 176; *Figure 3—figure supplement 1*). Furthermore, in vitro data shows that single positive T$^+$ and Sox2$^+$ cells can be derived from Sox2$^+$T$^+$ cells (*Tsakiridis and Wilson, 2015*), and cell clusters comprising 60–80% Sox2$^+$T$^+$ cells derived from pluripotent populations in vitro contribute to neural and mesoderm lineages when engrafted in the E8.5 embryo (*Gouti et al., 2014*). Taken together, this indicates that this marker combination is useful for prospectively identifying NMPs both in vivo and in vitro.

Interestingly, within the Sox2$^+$T$^+$ population, we show clear differences in levels of Sox2 and T (*Figure 5A*) and these appear to reflect greater likelihood to adopt neural and mesodermal fates respectively. Together with the observation that all L1-3 regions show equivalent potential on grafting to the NSB, this suggests that the levels of each protein may report on the cells' responses to environmental cues directing their fate.

### Functional equivalence between NMPs in the NSB and CLE

Our previous study identifying the NSB and CLE as NMP-containing regions (*Cambray and Wilson, 2007*) suggested that the NSB contained long-term NMPs, while the CLE contained shorter-lived progenitors. However the data presented here argue for a single, adaptable NMP type, since rostral CLE cells show equal potential to contribute to the CNH, as well as both neural and mesodermal fates over long axial distances, in the context of the NSB environment. Thus the differences in NSB and CLE fate do not result from intrinsic differences between the populations, but rather from the different environmental influences that they are exposed to in vivo.

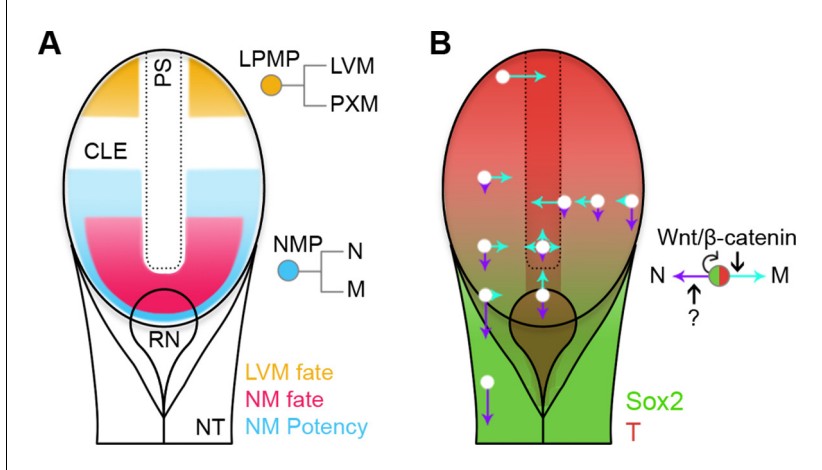

**Figure 11.** Position-dependent plasticity of primitive streak progenitors. (**A**) NMPs and LPMPs coexist in the caudal progenitor zone. The NSB and L1 regions show NM fate. NM potency is broader, encompassing L2–3 regions and coincides with a Sox2$^+$T$^+$ phenotype. LPMPs reside in the caudal-most embryo part; their potency to make PXM is somehow supressed. (**B**) Relative N vs M fate choices and trajectories for both progenitor populations are represented by directional purple and cyan arrows respectively, with length indicating proportions of cells entering each lineage. Background, relative expression of Sox2 (green) and T (red). Only NMPs require Wnt/β-catenin for mesoderm differentiation and maintenance.

## NMP plasticity may underpin axial morphogenesis

Fate mapping of the CLE and NSB (*Cambray and Wilson, 2007*) shows that only L1 and the NSB contribute significantly to the neurectoderm, and this almost always occurs along with mesoderm contribution, whereas a larger area, L2–3, contributes almost exclusively to mesoderm. This is consistent with single-cell labelling experiments HH4-9 chick embryos, which show that very few cells in the node region are fated solely for neural tube (*Selleck and Stern, 1991*). Clonal analysis in the mouse shows that medium-length mesoderm-only clones outnumber neuromesodermal clones (50 in mesoderm vs 8 NM; (*Tzouanacou et al., 2009*). Since the volume of differentiated paraxial mesoderm is always greater than that of the neurectoderm, the overproduction of mesoderm relative to neurectoderm may begin to be implemented in the primitive streak/tail bud region in the NMPs themselves. Together with the correlation between the mid-trunk expansion of NMPs and the size of the somites (*Tam, 1981*), this implies that regulation of NMP proliferation/survival and differentiation may begin the process of determining the final proportions of neurectoderm and mesoderm in the embryonic axis.

## Role of Wnt/β-catenin signalling in NMPs

Whilst it has been shown that Wnt/β-catenin signalling is required for mesoderm formation at the expense of neurectoderm, our results demonstrate that this bias represents a fate choice in NMPs themselves (*Figure 11B*). In zebrafish, downregulation of Wnt/β-catenin activity diverts cells fated for mesoderm towards neurectoderm, demonstrating that Wnt/β-catenin control of neural versus mesodermal fate choice is a vertebrate-wide phenomenon (*Martin and Kimelman, 2012*). The transcriptional downregulation of many Wnt pathway components in the emerging committed neurectoderm relative to the CLE is consistent with a role for Wnt downregulation in neural differentiation from NMPs (*Olivera-Martinez et al., 2014*).

In addition to the role of Wnt/β-catenin in NMP fate choice, we also uncover a novel function of Wnt/β-catenin signalling in the control of NMP cell numbers, since the expansion of Sox2$^+$T$^+$ cells between E8.5–9.5 depends on active Wnt/β-catenin (*Figure 9D and G*). A decline in *Wnt3a* expression in the elongating tail (*Cambray and Wilson, 2007*) occurs in tandem with the gradual loss of Sox2$^+$T$^+$ cells. Therefore it is likely that Wnt signalling is required for NMP maintenance. This suggests that the short axis of mutants with lowered Wnt3a levels (*Greco et al., 1996*; *Takada et al., 1994*) is due to a failure to maintain NMPs. Since, we show that T is dramatically downregulated in

the caudal epiblast when β-catenin is depleted, (*Figure 9G*), this raises the possibility that T is necessary for NMP maintenance. Interestingly, a recent study (*Denans et al., 2015*) shows that posterior *Hox* gene activation in chick leads to downregulation of Wnt signalling, and of *T* expression, slowing of cell ingression from the ectoderm layer and shortening of the PSM. The equivalent late axial elongation period in mouse begins around the trunk/tail transition (*Gomez et al., 2008*), when the number of NMPs begins to decline. Our data argues that the effects of posterior *Hox* genes on axial elongation operate in NMPs themselves via Wnt/β-catenin-mediated control of progenitor numbers (*Figure 11B*).

Despite strong evidence implicating Wnt/β-catenin in NMP fate choice and maintenance, several studies suggest that constitutive Wnt/β-catenin activity is not sufficient either to divert NMPs to mesoderm fates or to maintain NMPs. Providing the $T^+$ caudal region with a stabilised form of β-catenin, or overexpressing Wnt3a in $T^+$ or $Cdx2^+$ progenitors, results in an enlarged PSM domain (*Garriock et al., 2015*; *Aulehla et al., 2008*; *Jurberg et al., 2014*) but not an obvious reduction in $Sox2^+T^+$ NMPs in the CLE (*Garriock et al., 2015*). However in the converse experiment, when Wnt3a was deleted in the $T^+$ caudal region, it was not clear to what extent $Sox2^+T^+$ NMPs were affected (*Garriock et al., 2015*). Here we show that, when β-catenin is deleted, $Sox2^+T^+$ NMPs are significantly reduced in number, but are not completely eliminated from the caudal region. Taken together, these results show that elevated Wnt/β-catenin signalling alone is not enough to commit NMPs to a mesoderm fate, but its absence is sufficient to block mesoderm differentiation. This implicates additional signalling pathway(s) in driving NMPs towards mesoderm formation. Moreover, despite the requirement for Wnt signalling in expanding NMP numbers, at least some NMPs can tolerate both elevated and reduced levels of Wnt signalling, at least for short (24–48 hr) periods.

How can Wnt/β-catenin signalling have multiple distinct effects on NMPs during axis elongation? Nuclear β-catenin staining in the zebrafish tail bud has shown that canonical Wnt signalling is only moderately active in NMPs and high in early differentiating mesodermal cells (*Bouldin et al., 2015*). Since *Wnt3a* is expressed uniformly in the E8.5 caudal region (*Cambray and Wilson, 2007*; *Takada et al., 1994*), it is unlikely that localisation of Wnt3a itself can provide the specificity to direct mesoderm formation from NMPs. It is therefore possible that Wnt modulators might provide this signal. R-spondin3, an extracellular Wnt activator, is transcribed in the PSM (*Kazanskaya et al., 2004*) and has recently been shown to drive pluripotent cells towards skeletal muscle derivatives in vitro (*Chal et al., 2015*). Moreover, the ability to both maintain progenitors and permit their differentiation may lie in its coordination with other signalling pathways. Fgf signalling, as well as steroid biogenesis and chondroitin sulphate biosynthesis were identified as differentially regulated between NMPs and committed neural progenitors in chick (*Olivera-Martinez et al., 2014*), and it remains to be seen whether these play a role in coordinating the balance between neural and mesoderm differentiation in all vertebrates. There also appear to be temporal differences in cell responses to Wnt/β-catenin signalling. The data presented here and shown by Martin and Kimelman (*Martin and Kimelman, 2012*) shows that prospective neurectoderm can be re-fated to mesoderm. However, mouse embryos lacking Wnt3a or β-catenin in the primitive streak are still able to form rostral paraxial mesoderm, (*Aulehla et al., 2008*; *Takada et al., 1994*), suggesting the mesoderm-inducing signals acting on the early progenitors of paraxial mesoderm must be different from those at later times.

## Differential requirements for Wnt signalling in NMPs and LPMPs

Previous studies have not distinguished whether Wnt/β-catenin signalling acts specifically in NMPs or is a generic inducer of mesoderm fate. We show that unlike NMPs, LPMPs in the caudal PS do not require Wnt/β-catenin signalling for mesoderm differentiation. Moreover, our data suggests that LPMPs are less dependent on Wnt/β-catenin signalling for T maintenance, and that additional signalling could be an important regulator of this population. *Bmp4* and *Bmp7* are expressed in the caudal CLE (*Arkell and Beddington, 1997*; *Fujiwara et al., 2002*) and have been implicated in the upregulation of T during mesoderm differentiation from embryonic stem cells (*Suzuki et al., 2006*). Together with the inability of forced Sox2 expression to direct integration in the NT, our results indicate that LPMPs are regulated differently from NMPs. We also show that LPMPs can participate in PXM production, consistent with clonal data showing that mesoderm-restricted clones can contribute to both PXM and LVM (*Tzouanacou et al., 2009*). We did not observe an obvious bias in mediolateral distribution in PXM formed from LPMPs. However, it will be interesting to determine whether any downstream mesoderm lineages are affected in the absence of β-catenin. Our fate map shows

that L/St5 cells normally never contribute to the PXM, suggesting that LPMPs choose lateral meso-derm fates over paraxial ones based on their position.

### Interaction between NMPs and their neighbours

Here we show NMP numbers peak through the actions of Wnt/β-catenin signalling during trunk-to-tail transition. This role of β-catenin in NMP maintenance/expansion may be cell-autonomous, although it is also possible that it acts in neighbouring notochord progenitors. Wnt/β-catenin signalling is high in this region throughout axis elongation and is required in the notochord itself for notochord integrity and tail growth (*Ukita et al., 2009*). Abnormalities in the caudal notochord are also present in embryos depleted for β-catenin between E8.5–9.5 (*Figure 9Cd–f*), and the distance between the NMP area and the caudal notochord apparently increased (*Figure 9G*). Despite the structural aberrations in *βcatCKO* embryos, the level of T in the notochord is far less affected compared to other tail bud domains (*Figure 9Gd*), suggesting T expression is uncoupled from Wnt/β-catenin signalling in these cells. Notochordal cells are found close to NMPs in the NSB but not in the CLE, arguing against the idea that the presence of the notochord environment is absolutely essential for NMP identity. However, since the NMP pool is only maintained over long axial distances adjacent to the caudal notochord tip (*Figure 3C-F* and *Cambray and Wilson (2002)*; *McGrew et al. (2008)*), NMP maintenance may be indirectly dependent upon the notochord. This notochordal 'niche' could include extracellular matrix molecules, and secreted factors including Wnts. Therefore, Wnt/β-catenin signalling may act indirectly to maintain the NMP environment.

Similarly LPMPs are likely to interact with NMPs during the time of trunk-to-tail transition. Gdf11 binds Tgfbr1 and activates *Isl1* expression in the emerging LVM during trunk-to-tail transition and provides signals that induce hindlimb bud and cloaca formation (*Jurberg et al., 2013*). Ectopic *Isl1* activation in the whole tail bud, including NMPs, leads to complete loss of NM-derived trunk vertebrae and spinal cord (*Jurberg et al., 2013*), while loss of *Gdf11* function increases the interlimb area by 5 to 6 somites (*Jurberg et al., 2013*; *McPherron et al., 1999*). This suggests that Gdf11 acts via LPMPs to attenuate signals that expand the NMP population and, together with posterior *Hox* gene reduction of Wnt signalling (*Denans et al., 2015*; *Young et al., 2009*), control the timing of progenitor number reduction in the tail.

In conclusion, we have shown two separate progenitor types exist in the primitive streak, characterised by differential plasticity and commitment, as well as by the mechanisms by which they choose between alternative fates. Interestingly, both NMPs and LPMPs exhibit potency beyond their normal fates, yet this potential is intrinsically constrained in each case. Thus, besides serving as a benchmark for generation of differentiated cell types in vitro, these progenitor types can form a paradigm for testing how the interplay between intrinsic competence and extrinsic signals is set up and maintained during cell differentiation.

## Materials and methods

### Mouse strains, staging and husbandry

Wildtype, outbred MF1 and transgenic mice (*AGFP7* (*Gilchrist et al., 2003*), *βcatCKO* and *βcatCKO:sGFP* (*Tsakiridis et al., 2014*)), were maintained on a 12-hr-light/12 hr-dark cycle. F1 *βcatCKO:sGFP* embryos were obtained as shown in *Figure 9—figure supplement 1A*. For timed matings, noon on the day of finding a vaginal plug was designated E0.5.

### Dissection of embryos and tissue grafts

Dissection and in vitro culture of early somite stage embryos was performed as described (*Copp and Cockroft, 1990*). Embryos were cultured for 48 hr from E8.5 (2-5s) to E10.5 (30-35s) stage. Dissected PS regions shown in *Figure 1A*, corresponding to about 100–150 cells were dissected and grafted as described (*Cambray and Wilson, 2007*) with the exception that for CLE dissection, underlying mesoderm was manually dissected away from the ectoderm using hand-pulled solid glass needles. Since the L5 region is small, we could not exclude that a part of the midline (St5) was included in L5 grafts. Therefore, this region is termed L/St5. An overview of all grafting experiments is shown in *Figure 6—figure supplement 2A*. In homotopic grafts, L1-3 cells often colonised a short stretch of rostral somites unilaterally on the grafted side. In more caudal sections of the axis,

somite contribution in L1AT, L2-3 and L1-2med homotopic grafts were often bilateral (17/21 grafts). Therefore, it seems some of the grafted cells are able to encroach on the midline and produce mesoderm. In contrast, most L1AT and L2lat homotopic grafts (6/8 grafts) colonised the paraxial mesoderm to one side only (see also *Figure 6—figure supplement 4*). Importantly, unilateral PXM contribution might have arisen from pre-existing presomitic mesoderm that was co-grafted along with the NMPs of the CLE. This is likely not to be the case for the majority of the PXM contribution observed (see *Figure 6—source data 1*). We estimated this presomitic mesoderm contamination to be less than 20% of the total section number. In addition, bilateral PXM contribution is present in almost all homotopic CLE grafts, indicating that cells must have encountered the midline PS at some point in their history (*Nicolas et al., 1996*; *Eloy-Trinquet and Nicolas, 2002*), despite being grafted lateral to the streak (*Figure 6—source data 1*). The bilateral PXM contribution of most grafts also argues that most grafted cells are derived from CLE, which experiences a net movement towards the midline PS, while PSM progenitors would normally move away from it (*Psychoyos and Stern, 1996*; *Voiculescu et al., 2007*). Thus, the bulk of the PXM contribution in CLE grafts has most likely descended from NMPs rather than from pre-existing PSM progenitors. To verify grafted cell locations, embryos were imaged immediately following grafting in a fluorescent dissecting microscope (*Figure 7E*, *9A*, *10A and F*).

## Kidney capsule grafts

Kidney capsule grafts were performed as described previously (*Beddington, 1983*), and processed and scored as in (*Osorno et al., 2012*). Syngeneic kidney capsule grafts were performed in CBA or 129 mice. In E7.5 (late streak/early headfold) control grafts, 1 to 2 regions consisting of anterior or posterior halves of the embryonic region were transplanted, while up to 3 E8.5 (2–6 s) regions were grafted under the kidney capsule. In preliminary experiments, little difference in size between grafts of different numbers of regions was noted, nor was there a significant difference between the size of E8.5 tumours recovered at 4 and 6 weeks. Therefore, data from these grafts was pooled. Tumours were fixed in 4% paraformaldehyde for 1 to 7 days depending on size, processed and stained as described (*Bancroft and Gamble, 2002*). The tumour size was measured as the average surface area ( ± s.d.) in sections.

## Histology and scoring of graft contribution

Eight grafting experiment types were performed. In *series 1 (embryos 1.01–1.20)*, control homotopic grafts of L1, 2 or 3 tissue (without distinction between the three areas) were grafted to a recorded position in the L1 or L2-3 area of the CLE. In series 2 (*embryos 2.01–2.08*) either a more medial or lateral piece of the L1-2 epiblast was homotopically grafted. In *series 3 (embryos 3.01–3.04)*, L/St5 tissue was grafted to the same position of stage-matched embryos. In *series 4 (embryos 4.01–4.09)*, L1, 2 or 3 tissue (without distinction between the three areas) was grafted to the NSB. *Series 5 (embryos 5.01–5.18)* contained grafts of individual areas of L1, 2 or 3, divided either by their rostro-caudal or mediolateral position, or both, to the NSB. In *series 6 (embryos 6.01–6.21)*, defined L1, 2 or 3 regions were grafted to subregions within the NSB. The crown of the node was used as a landmark: Br grafts were positioned immediately rostral to the crown, Bc grafts immediately caudal to it, with grafts to Bm inserted at the crown itself. *Series 7 (embryos 7.01–7.03)* and *series 8 (embryos 8.01–8.09)* contained grafts of L/St5 to the primitive streak (St1-3) and the rostral part of the border (Br) respectively. All embryos were processed and scored as described below.

Embryos were photographed in wholemount and then fixed, mounted and sectioned in a vibratome (Series 1000, The Vibratome Company or VT1000M, Leica) at 50µm as described (*Cambray and Wilson, 2007*). Sections were counterstained with TO-PRO®-3 Iodide (Life Technologies, Carlsbad, CA) and visualised using an Olympus BX61 compound microscope with Optigrid confocal optics (Qioptiq, Waltham, MA). In the axis rostral to the tail bud, contribution of GFP$^+$ cells in each section to neural tissue, paraxial mesoderm, lateral/ventral mesoderm, axial mesoderm, endoderm or surface ectoderm was noted, along with any self-differentiated or non-integrated tissue. In the tail bud, contribution to the dorsal and ventral (notochordal) parts of the CNH and the tail bud mesoderm was scored. No graft-derived cells contributed to either endoderm or surface ectoderm. *Figures 6*, *7* and *10* display the scored sections containing integrated graft-

derived cells for each individual embryo, whereas *Figures 8* and *10* also show the contribution per tissue type for each graft cohort, shown as the percentage of the total number of scorable sections.

An additional six grafts were performed to test grafted cell identity using cryostat embedding, followed by immunohistochemistry (2x L1 homotopic, 2x L2-3 to Br, 1x L/St5 homotopic and 1x L/St5 to St1-3). These grafts had similar contribution patterns, but since not all sections were scored, they were not included in the above series. All L1-3 and L/St5 *βcatCKO:sGFP* to St1-3 grafted embryos were cryosectioned, examined for their contribution pattern, but not serially scored.

## Electroporation of donor tissue

L/St5 cells were micro-dissected from E8.5 (2–6 s) *AGFP7* embryos as described above. Several L/St5 tissue pieces (~100 cells each) were transferred to a glass electroporation well containing 1μg/mL CAG-Sox2-T2A-tdTomato plasmid in PBS (Sigma, St. Louis, MO). To obtain this plasmid, the Sox2 ORF was cloned into pPyCAGIP (*Chambers et al., 2003*), in which the IRES puromycin resistance cassette was exchanged for a nucleotide sequence corresponding to a T2A self-cleaving peptide (*Szymczak et al., 2004*) and a tdTomato fluorescent reporter. Two 1 mm, L-shaped, gold tip Genetrode electrodes (BTX Model 516, BTX, Holliston, MA) were placed at 3 mm distance and tissue pieces were electroporated using an Electro Square Porator ECM 830 (BTX) (30V, 5 pulses, 50 ms/pulse, 1s interval). The electroporated tissue pieces were subsequently grafted into wildtype stage-matched host embryos, as described above.

## Wholemount in situ hybridisation

Wholemount in situ hybridisation was performed as described previously (*Wilkinson, 1998*) except that proteinase K treatment was empirically adjusted according to embryo size and stage (time between 5–20 min). Dual-colour labelling of embryos was performed by simultaneously hybridising DIG-labelled *Sox2* (*Avilion, 2003*) and FITC-labelled *T* (*Herrmann, 1991*) probes. Antibody incubations and color reactions were carried out sequentially with BCIP/NBT and BCIP/INT (Roche, Switzerland). To inactivate the first AP enzyme, a heat inactivation step (at 65°C in MABT (100 mM maleic acid, 150 mM NaCl, 0.1% (v/v) Tween-20, pH 7.5) and an acid treatment (0.1 M glycine, pH 2.2) were carried out (both 30 min) after the first color reaction. After hybridisation, embryos were fixed, processed and sectioned as described below.

## Immunohistochemistry

Wholemount immunohistochemistry was performed as described previously (*Osorno et al., 2012*). Embryos were fixed in 4% PFA in PBS at 4°C for 2 hrs (HF) or overnight (>E8.5 embryos). Samples were costained against Sox2 (Abcam, United Kingdom; ab92494; 1:200) and T (R&D, Minneapolis, MN; AF2085; 1 μg/ml), followed by overnight incubation in PBS containing 4',6-diamidino-2-phenylindole (DAPI, Life Technologies). Confocal microscopy was performed after dehydration through a PBS/methanol series (10 min each), three 5 min washes in 100% methanol, clearing in 1:1 v/v methanol/BA:BB (2:1 benzyl alcohol:benzyl benzoate), and two washes in BA:BB. Selected embryos were embedded, sectioned and stained as described in (*Huang et al., 2012*). Primary antibodies (supplier, catalogue number and working concentration) were as follows: anti-Sox2 (Abcam; ab92494; 1:200, Santa Cruz, Dallas, TX; sc-17320; 1 mg/ml or Merck Millipore, Germany; AB5603; 5 mg/ml); anti-T (R&D; AF2085; 1 μg/ml or Santa Cruz; sc-17743; 1 mg/ml); anti-Foxa2 (Santa Cruz; sc-6554; 1 mg/ml or R&D; AF2400; 1 μg/ml); anti-GFP (Abcam; ab13970; 10 μg/ml); anti-Pax3 (DSHB, Iowa City, IA; 1:20); anti-Pax6 (DSHB; 1:20); anti-PDGFRβ (Abcam; ab32570; 1:100); anti-β-catenin (Sigma; C2206; 1:1000); anti-Active β-catenin (Millipore; 05–665, clone 8E7; 0.1 μg/ml); anti-N-cadherin (Sigma; C3865; 1:400).

## Sox2/T quantification and statistical analysis

Acquired confocal images were deconvoluted using Huygens software (SVI, The Netherlands) and saved as 8-bit tiff files. Due to the large size, images were cropped to get rid of most black areas, before the segmentation and co-expression analysis. For some samples, the brightness and/or contrast was increased in the blue channel (DAPI) to enhance weakly fluorescing nuclei, using ImageJ (imagej.nih.gov/ij/). The best parameters for nuclear segmentation were defined in cropped regions of greyscale DAPI z-stacks using FARSIGHT v0.4.5 software (www.farsight-toolkit.org/wiki/Nuclear_

Segmentation). These defined parameters were used on all original greyscale DAPI z-stacks and resulted in a segmented images (in the blue field), in which each nuclear volume could be identified by a unique greyscale-based identifier (*Al-Kofahi et al., 2010*). The segmented blue field (DAPI) was then overlaid with the red (Sox2) and far red (T) field. Single-cell fluorescence quantification and 3D rendering was performed using a software program, designed by GB (manuscript in preparation; for more information, see *Davies et al. (2013)*). The nuclear debris size (the minimum size below which a 'nucleus' identified by the segmentation algorithm is too small to be real and is therefore considered as debris in the analysis) was set as nuclear volume $\leq$2000 pixels. Initially, a threshold of detection was set for both fields using four internal controls for each individual sample. These controls were cropped and saved from the original z-stack to have (a) only Sox2$^+$ cells, (b) only T$^+$ cells, (c) both Sox2 or T single positive cells and (d) neither Sox2 nor T positivity within the region of interest (ROI). The threshold for Sox2 and T positivity was defined by comparing how accurately the obtained quantification result could predict the expected values for each ROI (pre-set as clearly separated or negative for both factors). The same software package and parameters were then used to analyse the sample. The statistical significance between developmental stages and Sox2/T fractions were tested with a standard unpaired Student's t-test, using Prism v6.0 software (GraphPad, La Jolla, CA). To correct for differences in image acquisition, especially those at greater optical depths (e.g. image acquisition was optimised to capture the Sox2$^+$T$^+$ population in E9.5 tails), an additional Welch correction was carried out to compare Sox2/T populations, though it did not alter the statistical significance; p-value<0.05 was considered significant (*, p-value<0.05; **, p-value<0.0001; ns, not significantly different).

## Manual segmentation controls

To calculate the margin of error that accompanies the use of any automated segmentation algorithm, we have compared our automated segmentation (using FARSIGHT v0.4.5 software) to a method involving manual identification of individual nuclei. For the latter, we designed a manual Region Of Interest (ROI) 'seeding' tool in the Icy software platform (*de Chaumont et al., 2012*) (plug-in developed by GB), allowing us to manually seed ellipse-shaped ROIs in the centre of each cell. After all cells were identified, the 'Active Contours' plug-in was used to grow the original seed to the correct nuclear volume (http://icy.bioimageanalysis.org/plugin/Active_Contours). This resulted in a segmented image that was processed as before. Six different embryonic regions were cropped from two different confocal stacks (in E8.5 and E10.5 samples) and their segmentation was compared for both methods. The degree of error at E8.5 is 15% (n = 4) and 18% in E10.5 samples (n = 2 regions). Our automated segmentation thus provides a reasonably accurate measure of the number of nuclei in any region of interest. Throughout this manuscript we therefore use the term 'cells' to refer to these segmented nuclear volumes for clarity.

## Image analysis

Whole embryo images were captured with a digital camera (Qimaging, Canada) attached to a Zeiss Stemi SV11 (Zeiss, Germany), Nikon AZ100 (Nikon, Japan) or Leica M165 FC microscope (Leica, Germany). Confocal images of wholemount immunostained embryos were acquired on a Leica SP8 or SPE, or Zeiss LSM510 Meta platform. An Olympus BX61 (Olympus, Japan) compound microscope with fluorescence optics was used to capture images of cryosections using Volocity software (Perkin Elmer, Waltham, MA). Additional Optigrid confocal optics (Qioptiq) were used to image vibratome sections. Image processing was done using Adobe Photoshop (Adobe Systems, San Jose, CA) or ImageJ software (imagej.nih.gov/ij/).

## Acknowledgements

We thank Sally Lowell, Anestis Tsakiridis, Lesley Forrester, Stavroula Skylaki and all lab members for critical reading of the manuscript; Carol Manson for help with animal maintenance; Suling Zhao for technical assistance.

# Additional information

## Funding

| Funder | Grant reference number | Author |
|---|---|---|
| Medical Research Council | MR/K011200 | Filip J Wymeersch<br>Yali Huang<br>Ron Wilkie<br>Frederick CK Wong<br>Valerie Wilson |
| Medical Research Council | G080297 | Filip J Wymeersch<br>Yali Huang<br>Ron Wilkie<br>Frederick CK Wong<br>Valerie Wilson |
| Wellcome Trust | WT100133 | Guillaume Blin |

The funders had no role in study design, data collection and interpretation, or the decision to submit the work for publication.

## Author contributions

FJW, Conceived and designed the experiments, Performed and analysed mouse embryo grafting and fate mapping experiments, Performed immunohistochemical staining, Quantification of Sox2/T populations and 3D reconstruction, Performed embryo electroporation and in situ hybridisation experiments, Wrote the manuscript; YH, Conceived and designed the experiments, Performed and analysed mouse embryo grafting and fate mapping experiments, Performed immunohistochemical staining, Performed and analysed Sox2/T immunostaining, Quantification of Sox2/T populations and 3D reconstruction, Contributed towards drafting the manuscript and approved the final version; GB, Quantification of Sox2/T populations and 3D reconstruction, Provided image analysis software, Contributed towards drafting the manuscript and approved the final version; NC, RW, Performed and analysed the kidney capsule grafts, Contributed towards drafting the manuscript and approved the final version; FCKW, Performed immunohistochemical staining, Supplied Sox2 expression plasmid, Contributed towards drafting the manuscript and approved the final version; VW, Conceived and designed the experiments, Performed and analysed the kidney capsule grafts, Performed and analysed mouse embryo grafting and fate mapping experiments, Supervised experiments, Wrote the manuscript

## Author ORCIDs

Guillaume Blin, http://orcid.org/0000-0002-9295-237X

## Ethics

Animal experimentation: Animal experiments were performed under the UK Home Office project license PPL60/4435, approved by the Animal Welfare and Ethical Review Panel of the MRC Centre for Regenerative Medicine and within the conditions of the Animals (Scientific Procedures) Act 1986.

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
