## [Decision Letter]

Thank you for submitting your work entitled "Position-dependent plasticity of distinct progenitor types in the primitive streak" for peer review at *eLife*. Your submission has been favorably evaluated by Janet Rossant (Senior editor) and three reviewers, one of whom, Marianne Bronner, is a member of our Board of Reviewing Editors.

The reviewers have discussed the reviews with one another and the Reviewing editor has drafted this decision to help you prepare a revised submission.

Summary:

The authors of this study uniquely apply intricate grafting techniques to test the fate and potency of wild type and transgenic epiblast and primitive streak (PS) cell populations in the early mouse embryo. This complements developmental genetic approaches to target such cell populations and uncovers several important new findings. The work carefully documents Sox2 and T expression and co-expression (a potential indicator of neuromesodermal progenitor identity) during development and correlates co-expression of these proteins with contribution to both neural and mesodermal lineages. They further show that CLE fates are plastic; for example, rostral CLE although fated to be largely neural can make mesoderm when transplanted medially into the node streak border (NSB). Furthermore, variability in fates from such grafting can be further mapped to the precise position within the NSB into which the grafts are placed – allowing the authors to conclude that there is no localized potency within the epiblast regions and that cell fates depend on environmental cues, which must be regionally distributed in the NSB. These are useful new data for the field, which help define the neuromesodermal progenitor cell population through development and elucidate how it is regulated.

The study is extended to investigate the requirement for Wnt/β-catenin signaling in primitive streak cell populations. The main conclusion from this latter work (that Wnt signaling promotes paraxial mesoderm fate) has, to some extent, already been drawn in recent studies using developmental genetic approaches, for example, from the work of Garriock 2015, Development and Jurberg et al. 2014, Dev Biol., as well as work (by the authors and others) on analogous cells derived from ES cells in vitro (Gouti et al. 2014, PLoS Biology; Turner et al. 2014, Development). However, the authors provide here further important detail of the involvement of Wnt/β-catenin signaling, demonstrating a requirement for expansion of the neuromesodermal progenitor pool during axis elongation and the Wnt-independent generation of paraxial mesoderm by more caudally located primitive streak cells.

Required revisions:

Overall, this study is well conducted and carefully analyzed. There are some omissions of other works and clarifications that should be addressed, as well as further validation of grafts and grafting techniques.

1) In the Introduction, the authors should mention fate maps of the primitive streak, including work in the chick (Psychoyos and Stern CD, Development 1996). In particular, this shows that different rostro-caudal regions of the PS contribute to medial and lateral somite, is there any indication that such organization exists in the mouse PS? It seems odd that this is not mentioned or discussed (see point 7 below).

2) Also in the Introduction (third paragraph), the authors argue that function of Wnt/β-catenin has not been assessed specifically in the CLE, referring to work using the T-Cre lines, which are active also active mesoderm progenitors. However, Jurberg et al. 2014 use not only T-Cre but a further line driving expression in the epiblast (as well as mesoderm) via a Cdx2 enhancer, which appears to allow an earlier test of function in the CLE. Importantly, this work reveals a severe phenotype that affects generation of neural and to some extent mesodermal progenitors, when Wnt3a is over-expressed. In addition, Wnt3a deletion using a T-Cre line (which is shown to drive expression in cells that make neural tissue as well as mesoderm) (Garriock et al. 2015) shows very a similar requirement for mesoderm differentiation to grafting experiments reported here. These works need to be mentioned further in the Introduction and in the Discussion their data and interpretation should be discussed in detail along with conclusions from the authors' experiments.

3) In the Results, did the authors assess the extent to which grafts of L1-3 contain Sox2/T co-expressing cells? – i.e. could they fix some grafts and do IHC to make sure that these grafts are accurate and define the extent to which they represent an NMP cell population based on Sox2/T co-expression? This is important for their claim that their grafting experiments with β-catenin mutant tissue represent a test of requirement specifically in NMPs (subsection “Wnt/β-catenin Signalling is required for Mesodermal Fate in NMPs”).

4) Related to point 3) above, from the fourth paragraph of the subsection "Fate in the Caudal Lateral Epiblast is highly regionalized”, there seem to be quite a few grafts of L1 and L2 epiblast that contain mesoderm – when they are grafted laterally these make dorsal neural tissue, which makes sense – and as the mesoderm also labeled is unilateral this is then considered to be contamination; when the grafts are placed medially and contribute bilaterally this is then used to suggest that at least some cells ingress through the streak to make mesoderm, but some mesoderm here must also be a contaminant. Although I can see their logic here this "contaminating" mesoderm rather undermines the grafting approach; to get a sense of the accuracy, can the authors state the number of grafts placed laterally that are mesoderm free? Can they improve this? Is it also possible that grafting into the epiblast layer (though the mesoderm?) leads to grafted epiblast cells being misplaced in the mesoderm? Is this apparent using a fluorescence-dissecting microscope?

5) In the second paragraph of the subsection “Neural versus Mesodermal Outcome is influenced by NSB Position”, a conclusion should be made here – do these data mean that there are distinct cell populations for neuro-mesodermal progenitors and notochord in NSB? There is some literature on notochord progenitor cell populations that could be discussed here (e.g. Yamanaka et al. 2007). This further relates to use of the term CNH and significance of the few Sox2/T positive cells in the region at the caudal tip of the notochord – as these seem unlikely to contribute to the notochord. I wonder why they (and the caudal notochord) are included in the CNH indicated in Figure 3; is this term (CNH) and the domain that includes the caudal notochord meaningful here?

6) From the data in the last paragraph of the subsection “Dorsolateral Bias in NT and CNH is reset upon Transplantation to the Midline NSB” – can the authors clarify whether there is any evidence that lateral CLE is derived from NMPs located more medially? If not where does it come from? The data presented suggest that the CLE moves medially to replace ingressed NMPs and must then become sox2/bra co-expressing, however, the neural tube gets longer so surely the "flow" of cells must be from NMPs to NPs to make the neural tube?

7) In the second paragraph of the subsection “The Caudal-most CLA exhibits Mesoderm-restricted Plasticity”, when β-catenin is deleted in L/st5, cells form PXM and LVM, but not neural – this seems an important result. Does this paraxial tissue ever express T, given other data indicating that Wnt/β-catenin signaling directly regulates T, can the authors comment on this? Is it possible to score for whether this different route to PXM reflects contribution to lateral rather than medial somite? Is it that NMP derived tissue contributes medial somite only?

8) In the first paragraph of the subsection “Interaction between NMPs and their Neighbours”, the authors need to clarify whether they are claiming that Wnt is required in notochord, which then provides other signals that promote /maintain NMPs or simply that notochord is a source of Wnts that act on the NMPs.

9) Images in Figure 3: signals for T and Sox2 are too weak and magnification too low to be informative of co-expression. Bc-I are out of focus compared to Ba Bb and others; can this be improved? Are these E8.5 observations different from Garriock et al. 2015?

---

## [Author Response]

*Required revisions: Overall, this study is well conducted and carefully analyzed. There are some omissions of other works and clarifications that should be addressed, as well as further validation of grafts and grafting techniques. 1) In the Introduction, the authors should mention fate maps of the primitive streak, including work in the chick (Psychoyos and Stern CD, Development 1996). In particular, this shows that different rostro-caudal regions of the PS contribute to medial and lateral somite, is there any indication that such organization exists in the mouse PS? It seems odd that this is not mentioned or discussed (see point 7 below).*

There seems to be a genuine interspecies difference between mouse and chick, albeit a subtle one. Fate mapping studies in chick embryos show that the medial part of the somites arises from a more rostral position in the PS than the lateral region (Psychoyos and Stern, Development 1996; Iimura and Pourquie, Nature 2006). Our equivalent mouse embryo grafts show that cells in the node-streak border contribute to the medial somites only, whereas elsewhere in the PS, cells contribute to the whole somite, not obviously excluding the medial region (Cambray and Wilson, Development 2007).

Our previous heterotopic grafts showed that cells fated exclusively for the medial somite in the NSB were re-fated to the whole somite on transplantation to the anterior PS. The converse was also true: grafting anterior PS cells to the NSB redirected cells towards the medial somite. While we believe that it was not necessary to revisit this issue in the present work, the re-fating of cells within the somite underlines the plasticity of mesoderm progenitors, and we agree it is relevant to mention in the context of the ability of lateral plate mesoderm progenitors to form somitic mesoderm on transplantation to the NSB and PS (see point 7).

We added a new paragraph to the Introduction:

“Fate mapping studies in early somite-stage mouse and chick embryos indicate a rostrocaudal organisation of mesoderm progenitors within the PS. […] However, the extent of mesoderm plasticity in the PS and CLE has not been fully investigated. In particular, it is not known whether axial, paraxial and lateral mesoderm progenitors in the PS region are committed to these fates.”

2) Also in the Introduction (third paragraph), the authors argue that function of Wnt/β-catenin has not been assessed specifically in the CLE, referring to work using the T-Cre lines, which are active also active mesoderm progenitors. However, Jurberg et al.

*2014 use not only T-Cre but a further line driving expression in the epiblast (as well as mesoderm) via a Cdx2 enhancer, which appears to allow an earlier test of function in the CLE. Importantly, this work reveals a severe phenotype that affects generation of neural and to some extent mesodermal progenitors, when Wnt3a is over-expressed. In addition, Wnt3a deletion using a T-Cre line (which is shown to drive expression in cells that make neural tissue as well as mesoderm) (Garriock et al. 2015) shows very a similar requirement for mesoderm differentiation to grafting experiments reported here. These works need to be mentioned further in the Introduction and in the Discussion their data and interpretation should be discussed in detail along with conclusions from the authors' experiments.*

We acknowledge that we should have dealt with these two papers more fully in the Introduction and Discussion. We feel, however, that there are flaws in the argument presented by Jurberg et al. (Dev. Biology 2014) that they are studying the role of Wnt signalling specifically in the epiblast. They compared two transgenic lines: a *T-Cre: del^*(ex3)/-*^*(a conditionally active β-catenin allele) and a *Cdx2P-Wnt3a* overexpressing transgene. The authors argue that the epiblast expression of the latter is responsible for its more severe phenotype. However, this is not a like-for-like comparison. The expression of activated β-catenin in descendants of T-expressing cells in *T-Cre:c del^(ex3)/-^*embryos or the difference between expressing a ligand, Wnt3a, and one of its downstream effectors, β-catenin, may account for aspects of the differences in phenotype. Moreover, there is expression of *T-Cre* in the epiblast, as shown by Perantoniet al. (Development 2007). We would therefore prefer not to emphasize the epiblast specificity of *Cdx2P-Wnt3a* in our discussion of this work.

Garriock et al. (Development 2015) compare a *T-CreER^T2^*versus a *Cited2-CreER^*T2*^*deletion of β-catenin, and further lineage-tracing shows expression of *T-creER^T2^*in the epiblast layer leads to the phenotypes studied. Although these arguments do seem valid to us, and allow the study of Wnt/β-catenin function in axial progenitors in general, none of them allow conclusions to be drawn specifically about NMPs in the CLE, and we do therefore believe we were right to claim that this has not yet been directly assayed.

Nevertheless, these studies are relevant and informative in the context of our work and we have added text to the Introduction and Discussion to deal more fully with them.

Added to the Introduction:

“More recently, lineage-tracing experiments showed that conditional deletion of Wnt3a or β-catenin in the T^+^ cell compartment leads to a switch of primitive streak progenitors towards a neural fate (Garriock, 2015). However, constitutive Wnt/β-catenin activity in the T^+^ cell compartment is not sufficient to divert all neural progenitors to mesoderm fates: providing cells in the caudal progenitor region with a stabilised form of β-catenin results in an enlarged PSM domain, but does not lead to loss of neural cell production (Aulehla, 2008; Jurberg, 2014). Moreover, enhanced β-catenin activity does not necessarily compromise the presence of NMPs in the CLE (Garriock, 2015). While these experiments point to an important role of Wnt signalling in axial progenitors, the promoters used do not specifically target NMPs. Grafting of precise NMP areas can provide a complementary approach that allows a direct assessment of the currently unresolved roles of Wnt signalling in NMPs and the caudal-most CLE.”

Added to the Discussion:

“Despite strong evidence implicating Wnt/ β-catenin in NMP fate choice and maintenance, several studies suggest that constitutive Wnt/ β-catenin activity is not sufficient either to divert NMPs to mesoderm fates or to maintain NMPs. Providing the T^+^ caudal region with a stabilised form of β-catenin, or overexpressing Wnt3a in T^+^ or Cdx2^+^ progenitors, results in an enlarged PSM domain (Aulehla, 2008; Jurberg, 2014; Garriock, 2015) but not an obvious reduction in Sox2^+^T^+^ NMPs in the CLE (Garriock, 2015). However in the converse experiment, when Wnt3a was deleted in the T^+^ caudal region, it was not clear to what extent Sox2^+^T^+^ NMPs were affected (Garriock, 2015). Here we show that, when β-catenin is deleted, Sox2^+^T^+^ NMPs are significantly reduced in number, but are not completely eliminated from the caudal region. Taken together, these results show that elevated Wnt/β-catenin signalling alone is not enough to commit NMPs to a mesoderm fate, but its absence is sufficient to block mesoderm differentiation. This implicates additional signalling pathway(s) in driving NMPs towards mesoderm formation. Moreover, despite the requirement for Wnt signalling in expanding NMP numbers, at least some NMPs can tolerate both elevated and reduced levels of Wnt signalling, at least for short (24-48 hour) periods.”

*3) In the Results, did the authors assess the extent to which grafts of L1-3 contain Sox2/T co-expressing cells?* –

*i.e. could they fix some grafts and do IHC to make sure that these grafts are accurate and define the extent to which they represent an NMP cell population based on Sox2/T co-expression? This is important for their claim that their grafting experiments with β-catenin mutant tissue represent a test of requirement specifically in NMPs (subsection “Wnt/β-catenin Signalling is required for Mesodermal Fate in NMPs”).*

To validate the accuracy of our dissection technique, we have added new data (Figure 6—figure supplement 1). This shows that dissected L1-3 regions (n=7) contain Sox2^+^T^+^ cells throughout, whereas L/St5 pieces (n=5) contain no double positive cells. This exactly matches the expected localisation of Sox2^+^T^+^ cells determined by wholemount immunohistochemistry, and we therefore believe that we have demonstrated that our dissection technique was accurate.

Added text to the Results:

“We analysed cell fate in the Sox2^+^T^+^ rostral CLE in both rostral-to-caudal (L1-3; Figure 6 and Figure 5) and medial-to-lateral axes (Lmed and Llat; Figure 6) and also in the Sox2^-^T^+^ caudal CLE and streak (L/St5; Figure 6). To check the accuracy of dissection, we performed immunohistochemistry on dissected L1-3 and L/St5 pieces.

While Sox2^+^T^+^ cells were abundant throughout L1-3 regions (n=7), no double positive cells were detected in L/St5 pieces (n=5; Figure 6—figure supplement 1). In general, homotopic grafts incorporated well in cultured embryos (37 incorporated/42 grafts performed; Figure 6—figure supplement 2 and Figure 6—figure supplement 3).”

We also added Figure 6—figure supplement 1.

4) Related to point 3) above, from the fourth paragraph of the subsection "Fate in the Caudal Lateral Epiblast is highly regionalized”, there seem to be quite a few grafts of L1 and L2 epiblast that contain mesoderm – when they are grafted laterally these make dorsal neural tissue, which makes sense – and as the mesoderm also labeled is unilateral this is then considered to be contamination; when the grafts are placed medially and contribute bilaterally this is then used to suggest that at least some cells ingress through the streak to make mesoderm, but some mesoderm here must also be a contaminant. Although I can see their logic here this "contaminating" mesoderm rather undermines the grafting approach; to get a sense of the accuracy, can the authors state the number of grafts placed laterally that are mesoderm free? Can they improve this? Is it also possible that grafting into the epiblast layer (though the mesoderm?) leads to grafted epiblast cells being misplaced in the mesoderm? Is this apparent using a fluorescence-dissecting microscope?

We thank the reviewers for spotting a lack of clarity on this important issue. We have added a supplementary file explaining our arguments that contaminating pre-existing mesoderm contamination cannot account for the majority of the mesoderm contribution seen in our grafts. We added [Supplementary-material SD1-data] and inserted the following passage in the Results:

“The failure of L1A grafts to contribute to the TB shows that these cells are en route for exit from the progenitor region, and the unilateral mesoderm contribution suggests that this mesoderm was already formed and carried alongside the CLE graft rather than ingressing through the PS. This is in agreement with previous reports that the presomitic mesoderm underlying the CLE contributes only to short stretches of axial tissue (estimated~6 somites) (Nicolas, 1996; Tam, 1986; Tam, 1988) (see also [Supplementary-material SD1-data]).”

We have adapted the text in the Methods, summarising the most important points and referring to the supplemental information:

“In contrast, most of L1AT and L2lat homotopic grafts (2/8 grafts) colonised the paraxial mesoderm to one side only (see also Figure 6—figure supplement 4). Importantly, unilateral PXM contribution might have arisen from pre-existing presomitic mesoderm that was co-grafted along with the NMPs of the CLE. This is likely not to be the case for the majority of the PXM contribution observed (see [Supplementary-material SD1-data]). […] Thus, the bulk of the PXM contribution in CLE grafts has most likely descended from NMPs rather than from pre-existing PSM progenitors.”

On the question of whether epiblast cells lodged in the mesoderm layer could directly differentiate to mesoderm, it is certainly true that the grafting technique lodges epiblast cells immediately adjacent to the mesoderm layer. However, the low mesoderm contribution of L1A and L2lat homotopic grafts argues that if this does occur, it does not constitute the bulk of mesoderm formed. Additionally, the high proportion of grafts containing bilateral mesoderm contribution argues that the grafted cells have moved towards the midline, a property of the epiblast and not the mesoderm, which moves away from the midline. We think this possibility is therefore unlikely, although it cannot be completely excluded.

We did not think that this question required additional text in the manuscript. We have no information to suggest that direct differentiation occurs, but even if it does, it would not alter our conclusions that NMPs make mesoderm, it only concerns whether the PS is required for this process.

5) In the second paragraph of the subsection “Neural versus Mesodermal Outcome is influenced by NSB Position”, a conclusion should be made here – do these data mean that there are distinct cell populations for neuro-mesodermal progenitors and notochord in NSB? There is some literature on notochord progenitor cell populations that could be discussed here (e.g. Yamanaka et al. 2007). This further relates to use of the term CNH and significance of the few Sox2/T positive cells in the region at the caudal tip of the notochord – as these seem unlikely to contribute to the notochord. I wonder why they (and the caudal notochord) are included in the CNH indicated in Figure 3; is this term (CNH) and the domain that includes the caudal notochord meaningful here?

This is a very interesting question. DiI labelling of the RN has shown that the ventral layer is exclusively fated for notochord. NSB grafts conducted in Cambray and Wilson (Development 2007) have shown us that this region can give rise to notochord, but we could not conclude whether this was derived from the ventral or dorsal layer. The results here verify that CLE-derived NMPs are not notochord progenitors. We cannot exclude that NSB-derived ones are, but since we have yet to detect any difference between NMPs at these two locations, it is likely that notochord progenitors are largely or completely segregated from the NMPs by the early somite stage. Yamanaka et al. (Dev. Cell 2007) made the interesting observation that new Noto^+^ notochord progenitors appear posterior to the 2-5 somite node, appearing to coalesce there to extend the notochord posteriorly. However, the origin of those cells is unclear, and our experiments suggest that they are not immediate descendants of the epiblast layer. The study by Brennan et al. (Genes Dev. 2002) indicates that the posterior notochord is derived from nodal positive cells in the mesoderm adjacent to the node at E8.5.

We added the following text to clarify the issue:

“Therefore, although they were able to enter the notochord domain, they were unable to differentiate correctly, suggesting that E8.5 NMPs are not notochord progenitors. This is consistent with fate mapping studies indicating that the posterior notochord is derived from nodal positive cells in the mesoderm adjacent to the node (Brennan et al., Genes Dev. 2002) and that convergent extension is the main driver of notochord extension at early somite stages (Yamanaka et al., Dev. Cell 2007). Together, these results suggest that from E8.5 onwards the notochord may elongate primarily as a result of rearrangement of pre-existing notochord progenitors rather than de novoaddition of cells from the epiblast layer.”

On the term CNH, we understand using ‘CNH’ to describe the location of late NMPs is probably not ideal, as it does include the caudal tip of the notochord, which is unlikely to contain NMPs. Indeed this region, which is analogous to the Xenopus CNH, may show some species differences, as the Xenopus CNH may contribute cells from the upper layer to the notochord (Gont et al., Development 1993). However we feel that the proper place to discuss this term in more detail would be a review on the topic, rather than a primary research paper, and that we should leave this term unchanged. It has the advantage of consistency with other publications (Cambray and Wilson, Development 2002; McGrew et al., Development 2008; Wilson et al., Development 2009).

*6) From the data in the last paragraph of the subsection “Dorsolateral Bias in NT and CNH is reset upon Transplantation to the Midline NSB”* – *can the authors clarify whether there is any evidence that lateral CLE is derived from NMPs located more medially? If not where does it come from? The data presented suggest that the CLE moves medially to replace ingressed NMPs and must then become sox2/bra co-expressing, however, the neural tube gets longer so surely the "flow" of cells must be from NMPs to NPs to make the neural tube?*

It is hard to imagine a medial-to-lateral flow of cells in the CLE during axis elongation, since, in addition to the data presented here showing a more medial location of cells in the caudal NT than in the more rostral parts of the axis (Figure 8).

A) We never saw dorsal neural tube contribution from homotopic NSB grafts (Cambray and Wilson, Development 2007) or heterotopic CLE to NSB grafts (this work).

B) Neuromesodermal clones show a general tendency to contribute to dorsal neural tube rostrally and ventral neural tube and mesoderm caudally (Tzouanacou et al., Dev. Cell 2009, Figure 5).

Thus we believe that there is a lateral-to-medial displacement of CLE cells. However, as the reviewers point out, there must be a supply of cells from the progenitor region to both the dorsal and ventral neural tube along the entire rostrocaudal axis. Figure 7 shows there is a dorsal-to-ventral shift in NT contribution as NMPs move towards the CNH (L1AT grafted cells contribute to more ventral NT stretches as they move into to the tail bud) and our working model (in Figure 10) shows that L1/2lat grafts are likely to have a higher propensity to generate neural tube over mesoderm derivatives It has been shown that the chick lateral CLE gives rise to more dorsal stretches of the neural tube (Catala et al., Development 1996), and our data is also consistent with the idea that the dorsoventral position at which a cell exits the CLE anteriorly predicts its eventual dorsoventral position. However, we do not have data that directly addresses the question of the source of dorsal neural tube over the length of the axis. Instead we have clarified the text slightly to emphasize that not all cells leave the CLE via the PS to become mesoderm.

We have adapted the text:

“CLE homotopic grafts predominantly contributed to lateral regions of the NT in the axis, but shifted towards the midline in the tail bud (Figure 6), Figure 6 indicating that there is a net displacement of at least some of the CLE cells are displaced towards the midline as axis elongation proceeds, perhaps to replace cells that have exited to the mesoderm via the midline PS.”

7) In the second paragraph of the subsection “The Caudal-most CLA exhibits Mesoderm-restricted Plasticity”, when β-catenin is deleted in L/st5, cells form PXM and LVM, but not neural – this seems an important result. Does this paraxial tissue ever express T, given other data indicating that Wnt/β-catenin signaling directly regulates T, can the authors comment on this? Is it possible to score for whether this different route to PXM reflects contribution to lateral rather than medial somite? Is it that NMP derived tissue contributes medial somite only?

We would like to rephrase the reviewers’ points as we understand the questions:

1) Does all prospective mesoderm (including future paraxial and future lateral mesoderm) express T, and are there differences in the regulation of T by Wnt/ β-catenin in these two populations?

Here we believe the reviewers are raising the interesting question of whether L/St5 cells form mesoderm independently of Wnt/β-catenin-driven T expression, even when forming paraxial mesoderm upon heterotopic grafting. While examining whether heterotopically grafted L/St5 tissue retains T expression to contribute to somites is beyond the scope of our study, we realised that our existing wholemount immunostained embryos could be re-analysed to address this question.

First, we examined whether both NMPs/nascent paraxial and LPMPs/nascent lateral mesoderm express T. We found that all these populations are positive for T expression, although expression in paraxial mesoderm is present up to a much more rostral level than that in lateral mesoderm.

To address whether T expression is depleted in descendants of E8.5 L/St5 after β-catenin deletion, we analysed the relative levels of T protein in β-catenin CKO and WT embryos and plotted them on the 3D reconstructed scaffold (shown in Figure 8 and Figure 8—figure supplement 3). Our findings are summarised below:

Spatial differences in T protein expression:

Quantitation of T expression, as well as that of Sox2, has confirmed our initial impression of a gradient of increasing T expression towards the caudal end of the PS, and an opposing gradient of Sox2 expression towards the rostral end. Interestingly, we can now definitively say that Sox2/T coexpressing cells express low levels of both proteins, and that LPMPs express higher levels of T than NMPs. Moreover, the levels of T expression in LPMPs overlap with those in the notochord, where most of the high-expressing cells are found.

Effect of β-catenin loss on T expression in NMPs and LPMPs:

We considered that the descendants of cells in the PS between E8.5 and E9.5 would reflect the effect of depleting β-catenin upon T protein from E8.5. These will be found in the emerging PXM and LVM, as well as in the PS itself. While we do not have a definitive fate map of the E9.5 PS, LVM progenitors should be located caudally to NMPs, which are marked by T/Sox2 coexpression. We find that T protein levels are significantly reduced in all but the highest-expressing cells in β-catenin CKO embryos. This reduction is found predominantly in the area containing NMPs and the emerging paraxial mesoderm. The area posterior to T/Sox2 coexpressing cells (putative LPMPs) and the emerging LVM, as well as the notochord, are not as severely affected. This data therefore supports a Wnt/ß-catenin independent mechanism to regulate T expression in LPMPs.

2) Are L/St5 (wildtype or β-catenin deficient) cells more likely to contribute to the lateral somite?

We added more text to the Introduction (see point 1) to explain that NMPs in the CLE contribute to both medial and lateral somite, while those in the NSB contribute only to medial somite.

We have observed L/St5 descendants in both medial and lateral parts of the somite, in both WT and β-catenin CKO embryos. Therefore, qualitatively, there appears to be no block to L/St5 cell entry to any part of the somite. Furthermore, depletion of β-catenin does not appear to preclude entry of L/St5 cells to the somite. Furthermore, *βcatCKO* cells derived from NMPs in the rostral part of the axis, where β-catenin would have been depleted after NMPs had entered the paraxial mesoderm lineage, are capable of differentiation as both medial and lateral regions of the somite (Figure 9). This supports the conclusion that Wnt/β-catenin is required specifically in NMPs to form mesoderm, rather than affecting the differentiation of paraxial mesoderm.

To answer the above two questions, we have added the following data:

Figure 5 showing the spatial differences in T and Sox2 protein during PS-to-TB transition;

Extra data in Figure 9 to show that β-catenin depletion affects NMPs and their derivatives differentially from LPMPs and their derivatives;

Two supplemental figures: Figure 9—figure supplement 3, showing additional supporting data on the quantitative image analysis, and Figure 10—figure supplement 2, showing the mediolateral contribution of grafted cells to the somites.

We added/edited figures and changed figure legends of Figure 5, Figure 9, Figure 9—figure supplement 3 and Figure 10—figure supplement 2.

We added text in the Results:

“Thus the number of putative NMPs peaks during mid-trunk formation. […] Strikingly, Sox2^+^T^+^ cells were excluded from high Sox2 and high T expressing regions (Figure 5).”

“Compared to wildtype controls, the notochord extended further caudally than in wildtype embryos […] the number of Sox2^+^T^+^ cells in E9.5 β-catenin-deleted embryos was not significantly different from untreated E8.5 samples, indicating a failure to expand NMP numbers (Figure 9).

“To determine the relative effects of β-catenin depletion on the different progenitor populations in the PS, we compared the different levels of T expression in WT and 4-OHT-treated *βcatCKO* embryos. […] Thus, Wnt signalling is required for NMP expansion, at le ast in part through the maintenance of T. Moreover, our data suggests the maintenance of T in LPMPs and their descendants is less dependent on β-catenin. Therefore, LPMPs may represent an alternative, β-catenin-independent route towards mesoderm formation.”

“However contribution to the TBM was limited (Figure 10 and Figure 10—figure supplement 1). We did not observe obvious differences in the mediolateral distribution of PXM descendants in any of the L/St5 grafts (Figure 10—figure supplement 2).”

We added text in the Discussion:

“Importantly, however, *T* transcripts extend more laterally in the epiblast than protein (Figure 2; (Wilson et al., Development 1995)), suggesting that Llat may already be poised to accumulate T protein. […] Finally, Sox2^+^T^+^ cells first appear at late neural plate stage, several hours before pluripotency is lost in the epiblast (Osorno et al., Development 2012).”

“Interestingly, within the Sox2^+^T^+^ population, we show clear differences in levels of Sox2 and T (Figure 5) and these appear to reflect greater likelihood to adopt neural and mesodermal fates respectively.

“In addition to the role of Wnt/β-catenin in NMP fate choice, we also uncover a novel function of Wnt/β-catenin signalling in the control of NMP cell numbers, since the expansion of Sox2^+^T^+^ cells between E8.5-9.5 depends on active Wnt/β-catenin (Figure 9). […]Interestingly, a recent study (Denans, et al, eLIFE 2015) shows that posterior *Hox* gene activation in chick leads to downregulation of Wnt signalling, and of T expression, slowing of cell ingression from the ectoderm layer and shortening of the PSM.”

“We show that unlike NMPs, LPMPs in the caudal PS do not require Wnt/β-catenin signalling for mesoderm differentiation. Moreover, our data suggests that LPMPs are less dependent on Wnt/β-catenin signalling for T maintenance, and that additional signalling could be an important regulator of this population. […]Our fate map shows that L/St5 cells normally never contribute to the PXM, suggesting that LPMPs choose lateral mesoderm fates over paraxial ones based on their position.”

*8) In the first paragraph of the subsection “Interaction between NMPs and their Neighbours”, the authors need to clarify whether they are claiming that Wnt is required in notochord, which then provides other signals that promote /maintain NMPs or simply that notochord is a source of Wnts that act on the NMPs.*

We rephrased the following passage:

“Here we show NMP numbers peak through the actions of Wnt/β-catenin signalling during trunk-to-tail transition. This role of β-catenin in NMP maintenance/expansion may be cell-autonomous, although it is also possible that it acts in neighbouring notochord progenitors. […] However, since the NMP pool is only maintained over long axial distances adjacent to the caudal notochord tip (Figure 3 and (Cambray and Wilson, Development, 2002; McGrew et al., Development 2008)), NMP maintenance may be indirectly dependent upon the notochord. This notochordal ‘niche’ could include extracellular matrix molecules, and secreted factors including Wnts. Therefore, Wnt/β-catenin signalling may act indirectly to maintain the NMP environment.”

9) Images in Figure 3: signals for T and Sox2 are too weak and magnification too low to be informative of co-expression. Bc-I are out of focus compared to Ba Bb and others; can this be improved? Are these E8.5 observations different from Garriock et al. 2015?

Unfortunately, the PDF file given to the reviewers has lost some of the detail that was present in the separate TIFF figure files, likely due to several steps of re-sizing and file-format conversion (figures were submitted as a PDF in the main text). We would ask the reviewers to check the original TIFF format files and trust that sufficient image quality can be preserved if accepted for publication.